# Structures of Marburgvirus glycoprotein and its complex with NPC1 receptor

Gang Ye[1,2,5✉], Fan Bu[1,2,5], Hailey Turner-Hubbard[1,2], Morgan Herbst[1,2], Lanying Du[3], Ge Yang[4], Bin Liu[4✉] & Fang Li[1,2✉]

Marburgviruses (MBVs) cause severe haemorrhagic fever with higher fatality rates than Ebola virus (EBOV)[1–4]. Here we show that the MBV glycoprotein (GP) mediates viral entry more efficiently than EBOV GP. Using cryo-EM, we determined structures of MBV GP in three states: (1) unbound; (2) bound to its endosomal receptor NPC1; and (3) complexed with a neutralizing nanobody. The glycan cap shields the receptor-binding site from NPC1 but only partially from the nanobody, enabling limited immune evasion. After glycan cap cleavage, NPC1 binds to MBV GP in a distinct orientation compared with EBOV GP, providing an additional anchor and enhancing receptor affinity. NPC1 engagement also induces substantial conformational changes in MBV GP, probably facilitating membrane fusion. Furthermore, MBV GP is susceptible to the neutralizing nanobody, which mimics NPC1 at the receptor-binding site. Together, our findings reveal MBV GP as a highly efficient entry mediator and suggest structural mechanisms that may contribute to its enhanced entry efficiency.

MBVs, members of the Marburgvirus genus within the *Filoviridae* family, cause severe haemorrhagic fever in humans and non-human primates[3,4]. Since their discovery in 1967, 18 MBV outbreaks have been reported, most recently in 2024[1]. The two known MBV species, Marburg virus (MARV) and Ravn virus (RAVV), are closely related and often not distinguished in outbreak reports. MBVs have been isolated from bats[5], confirming bats as their natural reservoir and suggesting long-term co-existence with human populations. The average case fatality rate of MBV infection is 73% (409 deaths among 563 reported human cases), markedly higher than that of EBOV, a member of the Ebolavirus genus, which averages 44% (14,881 deaths among 33,820 cases)[1,2]. Although two antibody-based therapies and one vaccine have been approved for EBOV[6,7], no licensed therapeutics or vaccines exist for MBVs. Understanding the causes of the high lethality of MBV and developing effective countermeasures are critical for pandemic preparedness and global health security.

Viral entry into host cells is a key determinant of infectivity and pathogenesis, and a major target for neutralizing antibodies[8,9]. The EBOV GP, which mediates entry, has been extensively studied[10,11]. On the viral surface, GP exists in a trimeric pre-fusion state composed of three copies each of the receptor-binding subunit GP1 and the membrane-fusion subunit GP2 (ref. 12). A defining feature of EBOV GP is that its receptor-binding site (RBS) is shielded by a glycan cap and a mucin-like domain (MLD), blocking antibody access and promoting immune evasion[13]. During entry, GP1 engages host factors at the cell surface to trigger endocytosis[14]. Inside the endosome, proteases remove the glycan cap and MLD[15], exposing the RBS and generating a cleaved form of GP (GPcl) that binds its receptor, Niemann–Pick C1 (NPC1)[10,11,16]. The crystal structure of EBOV GPcl bound to domain C of NPC1 (NPC1-C) shows that NPC1-C inserts two loops into a hydrophobic cavity at the top of GPcl to engage the RBS[10]. NPC1 binding then drives GP2 to transition into its post-fusion conformation, mediating membrane fusion[17]. Thus, EBOV entry depends on three essential steps: glycan cap removal, NPC1 binding and NPC1-induced conformational rearrangements of GP.

Compared with EBOV GP, MBV GP is much less characterized. Like EBOV, MBV GP uses NPC1 as its endosomal receptor[18]. To date, only two crystal structures of MBV GP have been reported, both in complex with neutralizing antibodies[13,19]. The structures of MBV GP alone or bound to NPC1 remain unknown, leaving major gaps in our understanding of its function. Neither the binding affinity of MBV GP for NPC1 nor the structural changes induced by NPC1 have, to our knowledge, yet been defined. Unlike EBOV GP, where the RBS is shielded, neutralizing antibodies can directly target the RBS of MBV GP[13,19]. This observation has led to the suggestion that the MBV GP glycan cap is flexible, permitting antibody access and challenging its presumed role in immune evasion[13,19]. Of note, the efficiency of MBV GP in mediating viral entry has not been investigated, despite its likely contribution to the high lethality of MBV infections. In this study, we compared the entry efficiencies mediated by MBV and EBOV GPs. We determined cryo-electron microscopy (cryo-EM) structures of MBV GP in three distinct states: (1) unbound, (2) bound to human NPC1-C, (3) and bound to a neutralizing nanobody (a single-domain antibody from camelids[20–24]). To complement these structural analyses, we performed biochemical assays. Together, our findings establish MBV GP as a highly efficient mediator of viral entry and provide new structural insights into its receptor recognition, cell entry and immune evasion, while also informing potential therapeutic strategies against MBV infection.

[1]Department of Pharmacology, University of Minnesota Medical School, Minneapolis, MN, USA. [2]Center for Emerging Viruses, University of Minnesota, Minneapolis, MN, USA. [3]Institute for Biomedical Sciences, Georgia State University, Atlanta, GA, USA. [4]Hormel Institute, University of Minnesota, Austin, MN, USA. [5]These authors contributed equally: Gang Ye, Fan Bu. ✉e-mail: yeg@umn.edu; liu00794@umn.edu; lifang@umn.edu

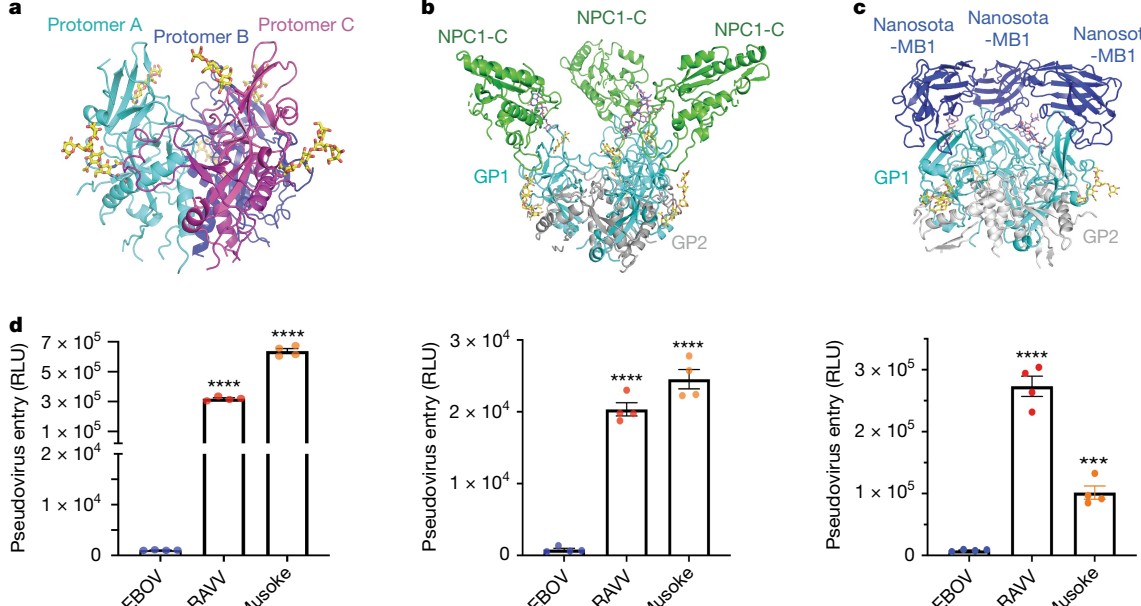

**Fig. 1 | Overall structure of MBV GP and its role in viral entry. a**, Trimeric structure of RAVV GPcl, with protomers shown in cyan, blue, and magenta; each protomer contains two N-linked glycans (yellow). **b**, Structure of trimeric RAVV GPcl bound to its endosomal receptor NPC1-C. GP1 and GP2 are shown in cyan and grey, respectively, and NPC1-C molecules are shown in green. **c**, Structure of RAVV GP-ΔM in complex with the neutralizing nanobody Nanosota-MB1 (blue). **d**, RAVV GP-mediated entry into human Huh7 cells (left), HUVECs (middle) and THP-1-derived macrophages (right). Retroviruses pseudotyped with wild-type, full-length, untagged GP from RAVV, MARV Musoke or EBOV were used to infect target cells. Entry signals were normalized to GP expression levels in each pseudovirus using a double normalization procedure (see Extended Data Fig. 4). Data are presented as mean ± s.e.m. ($n = 4$ biologically independent samples). Statistical differences between EBOV GP and MBV GPs (RAVV and Musoke) were determined using two-tailed, unpaired Student's $t$-tests, with $P$ values as follows: for Huh7 cells, EBOV versus RAVV $P < 0.0001$ and EBOV versus Musoke $P < 0.0001$; for HUVECs, EBOV versus RAVV $P < 0.0001$ and EBOV versus Musoke $P < 0.0001$; for THP-1-derived macrophages, EBOV versus RAVV $P < 0.0001$ and EBOV versus Musoke $P = 0.0001$. The asterisks above the bars indicate statistical significance (****$P < 0.0001$ and ***$P < 0.001$). RLU, relative light unit.

## Overall structure and function of MBV GP

To investigate the structure and function of MBV GP, we expressed and purified GP ectodomains from both MARV and RAVV. The RAVV GP ectodomain exhibited substantially higher expression yield and stability than MARV GP ectodomains from the Musoke and Angola strains, making it the preferred candidate for cryo-EM analysis. We first generated a truncated RAVV GP ectodomain lacking the MLD, referred to as GP-ΔM. Trypsin digestion was then used to remove the glycan cap, yielding GPcl. We performed cryo-EM analysis on RAVV GPcl, and the final structure was resolved at 3.17 Å (Fig. 1a, Extended Data Table 1 and Extended Data Fig. 1). We next expressed human NPC1-C and determined the cryo-EM structure of RAVV GPcl in complex with NPC1-C at 3.53 Å resolution (Fig. 1b, Extended Data Table 1 and Extended Data Fig. 2). To our knowledge, these represent the first resolved structures of MBV GP alone and in complex with NPC1.

Furthermore, we immunized an alpaca with RAVV GP-ΔM, isolated peripheral blood mononuclear cells and constructed a phage display library representing the nanobody repertoire of the alpaca[25]. Using RAVV GPcl as bait, we screened the library and identified a nanobody that effectively neutralized RAVV pseudovirus entry (see below). We named this nanobody Nanosota-MB1. We then determined the cryo-EM structure of RAVV GP-ΔM in complex with Nanosota-MB1 at 2.98 Å resolution (Fig. 1c, Extended Data Table 1 and Extended Data Fig. 3). To our knowledge, Nanosota-MB1 is the first neutralizing nanobody identified against MBVs, and this represents the first resolved structure of MBV GP in complex with a neutralizing nanobody.

In addition to structural analyses, we evaluated the ability of MBV GP to mediate viral entry in comparison with EBOV GP. We performed pseudovirus entry assays using retroviruses pseudotyped with MBV GP (RAVV or Musoke) to infect three human cell types: Huh7 (human hepatoma cells), primary human vascular endothelial cells (HUVECs) and macrophages differentiated from THP-1 (human monocytic leukaemia cells), all major cellular targets of filoviruses (Fig. 1d and Extended Data Fig. 4). EBOV pseudoviruses were included for direct comparison.

A long-standing challenge has been the lack of a reliable method to directly compare viral entry mediated by different filovirus GPs when normalized to equivalent expression levels. This difficulty arises because no neutralizing sera or antibodies recognize EBOV and MBV GPs equivalently, and the use of a common artificial tag risks interfering with entry. To address this, we developed a novel double-normalization strategy (Extended Data Fig. 4 and Supplementary Fig. 1). Specifically, we generated two versions of each GP: wild type and C-terminally tagged with C9. We then measured entry efficiency for six pseudoviruses: EBOV, RAVV and Musoke, each with or without the C9 tag. In the first step, wild-type EBOV, RAVV and Musoke pseudoviruses were normalized to their respective C9-tagged counterparts using neutralizing sera specific to each GP. In the second step, C9-tagged RAVV and Musoke pseudoviruses were further normalized to C9-tagged EBOV pseudoviruses using anti-C9 antibodies. This two-step procedure ensured that all six pseudoviruses were standardized to equivalent GP expression, enabling direct comparison of viral entry mediated by the three wild-type GPs.

The results showed that both wild-type RAVV and Musoke pseudoviruses entered all three human cell types far more efficiently than wild-type EBOV pseudoviruses (by more than 300-fold, 25-fold and 12-fold in the respective cell types), demonstrating that MBV GP is substantially more effective at mediating viral entry than EBOV GP (Fig. 1d). Although these data support higher entry efficiency mediated by MBV GPs under conditions of matched total GP expression, our assay cannot exclude differences in the proportion of functional versus misfolded GP produced in HEK293T cells and incorporated into pseudovirus particles.

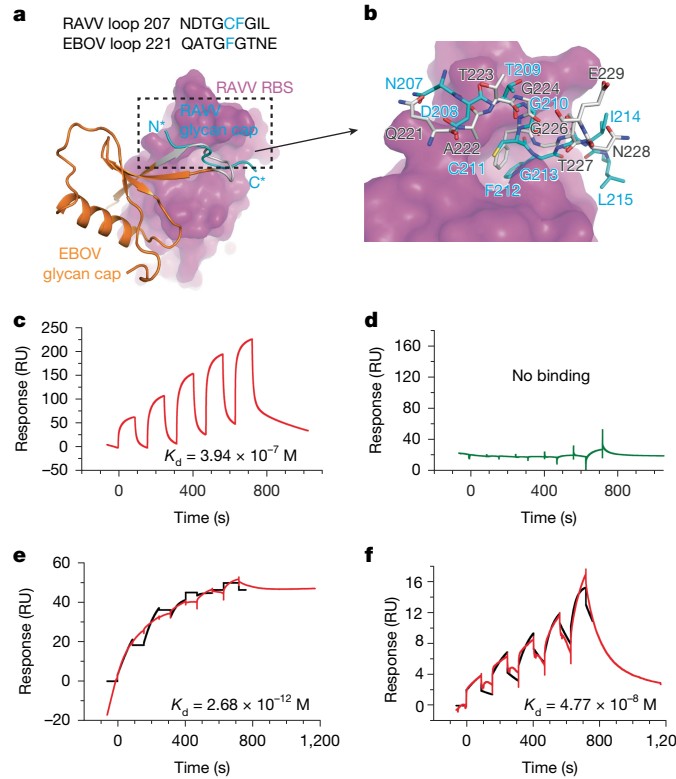

**a**

RAVV loop 207  NDTG**CF**GIL
EBOV loop 221  QATG**F**GTNE

RAVV RBS

RAVV
glycan cap

N*

C*

EBOV
glycan cap

**b**

T223  T209  E229
N207  G224
D203  G210
G226  I214
Q221  A222
C211  T227  N228
F212  G213  L215

**c**

Response (RU)
250
200
150
100
50
0
−50

0  400  800
Time (s)

$K_d = 3.94 \times 10^{-7}$ M

**d**

Response (RU)
160
120
80
40
0

No binding

0  400  800
Time (s)

**e**

Response (RU)
60
40
20
0
−20

0  400  800  1,200
Time (s)

$K_d = 2.68 \times 10^{-12}$ M

**f**

Response (RU)
16
12
8
4
0

0  400  800  1,200
Time (s)

$K_d = 4.77 \times 10^{-8}$ M

**Fig. 2 | Structure and function of the MBV GP glycan cap. a**, Comparison of the RBS-interacting glycan cap loops in RAVV GPcl and EBOV GP-ΔM (bound to a glycan cap-targeting nanobody; Protein Data Bank (PDB) ID: 9BSU). The RAVV RBS is shown as a pink surface, whereas the glycan cap loops from RAVV and EBOV are displayed as cyan and grey ribbons, respectively. N* and C* indicate the termini of the RAVV glycan cap loop. The amino acid sequences of the two loops − loop 207 (RAVV) and loop 221 (EBOV) − are also shown. **b**, Detailed structural comparison of the glycan cap loops in RAVV GPcl (cyan) and nanobody-bound EBOV GP-ΔM (grey). **c**–**f**, SPR analysis. NPC1-C binds to RAVV GPcl (**c**) but not to RAVV GP-ΔM (**d**), whereas Nanosota-MB1 binds to RAVV GPcl with ultra-high affinity (**e**) and to RAVV GP-ΔM with lower, but still strong, affinity (**f**). For NPC1-C binding to RAVV GPcl, SPR was performed in three independent replicates; mean $K_d$ values from all replicates and a representative binding curve are shown in **c** (see Supplementary Fig. 2 for all curves). RU, response unit.

## MBV GP glycan cap

The overall cryo-EM structure of RAVV GPcl closely resembles the two previously reported crystal structures of RAVV GP bound to human antibodies[13,19], featuring three GP1 subunits positioned atop a trimeric GP2 stalk (Fig. 1a and Extended Data Fig. 5a). A surprising finding in our structure is the presence of a nine-residue loop, loop 207 (Asn207–Leu215), from the glycan cap bound to the RBS (Fig. 2a and Extended Data Fig. 5b). This was unexpected, as SDS−PAGE analysis confirmed that the glycan cap was completely cleaved in RAVV GPcl. In previous crystal structures of RAVV GP-ΔM−antibody and RAVV GPcl−antibody complexes, no remnants of the glycan cap were observed[13,19]. By contrast, in our recently determined structure of EBOV GP-ΔM bound to a glycan cap-targeting nanobody, the glycan cap was visible and engaged with the RBS[25] (Fig. 2a,b). Structural alignment of apo RAVV GPcl with nanobody-bound EBOV GP-ΔM identified loop 221 in EBOV GP as the structural counterpart of loop 207 in RAVV GPcl, although the two loops differ markedly in sequence and secondary structure (Fig. 2a,b). In EBOV GP, loop 221 folds back to form an anti-parallel β-sheet, whereas in RAVV GPcl, loop 207 extends across the RBS. Despite these differences, both loops occupy a conserved hydrophobic pocket in the EBOV and RAVV RBSs, mediated by Cys211 and Phe212 in RAVV GP and Phe225 in EBOV GP. Contrary to previous conclusions that the RAVV glycan

cap is entirely flexible[13,19], our findings suggest that its flexibility is constrained by association with the RBS in the ligand-free state.

In our RAVV GPcl−NPC1-C and GP-ΔM−Nanosota-MB1 complex structures, the glycan cap loop was absent, displaced by the receptor and nanobody, respectively (Fig. 1b,c). This indicates that the glycan cap regulates ligand access to the RBS. To test this, we measured the binding affinities of RAVV GP (GP-ΔM and GPcl) for NPC1-C and Nanosota-MB1 using surface plasmon resonance (SPR). RAVV GPcl bound NPC1-C with a dissociation constant ($K_d$) of $3.94 \times 10^{-7}$ M, whereas GP-ΔM showed no detectable binding (Fig. 2c,d). Nanosota-MB1 bound GP-ΔM with a $K_d$ of $4.77 \times 10^{-8}$ M, indicating strong binding but substantially weaker than its ultra-tight interaction with GPcl ($K_d = 2.68 \times 10^{-12}$ M; Fig. 2e,f). These results confirm that the MBV GP glycan cap possesses partial flexibility, fully blocking NPC1 binding, whereas only partially restricting the binding of the more tightly interacting nanobody.

## MBV GP−NPC1 interactions

The structure of the RAVV GPcl−NPC1-C complex revealed that three NPC1-C molecules bind to the trimeric RAVV GPcl, with each NPC1-C engaging one RBS (Fig. 1b and Extended Data Fig. 5c). When we super-imposed the RAVV GPcl−NPC1-C and EBOV GPcl−NPC1-C complexes, NPC1-C was rotated by 39.4° in the RAVV complex relative to its orientation in the EBOV complex (Fig. 3a,b). To understand this difference, we analysed the RBS-binding loops of NPC1-C. Whereas NPC1-C uses two protruding loops (loop 1, residues 418–425; loop 2, residues 498–508) to bind the EBOV RBS, it uses three loops, including an additional loop 3 (residues 436–437), to bind the RAVV RBS (Fig. 3c,d). Loops 1 and 2 undergo major conformational changes when transitioning from EBOV to RAVV binding, whereas loop 3 provides an extra anchor that stabilizes the rotated orientation of NPC1-C in the RAVV complex (Fig. 3c,d). Consistent with these structural differences, affinity measurements showed that RAVV GPcl binds to NPC1-C approximately 11-fold more strongly than EBOV GPcl ($K_d$ values of 394 nM and 4.34 μM, respectively; Fig. 3b,e). Thus, compared with EBOV GPcl, RAVV GPcl engages NPC1-C through a distinct binding mode and with substantially higher affinity.

We next analysed the detailed interactions between the RAVV RBS and NPC1-C. Compared with the EBOV−NPC1-C interface, the RAVV−NPC1-C interface exhibits substantial differences. In total, 21 residues from the RAVV RBS directly contact 17 residues of NPC1-C (Extended Data Figs. 6 and 7). Of these, only 8 of the 21 RAVV residues are conserved in the EBOV RBS, whereas 11 of the 17 NPC1-C residues are shared between the two interfaces. A key difference is that Cys121 and Cys147 in the EBOV RBS − linked by a disulfide bond − are replaced by Leu105 and His131 in the RAVV RBS (Extended Data Fig. 8a,b). These substitutions shift the RAVV RBS closer to loop 3 of NPC1-C, thereby establishing loop 3 contacts (Extended Data Fig. 7c). These contacts specifically involve Asp436 and Ile437 of NPC1-C (Extended Data Fig. 7c). Alanine substitutions at these two positions markedly reduced RAVV GPcl-binding affinity (Extended Data Fig. 7d and Supplementary Fig. 2), confirming that loop 3 contributes to RAVV recognition. By contrast, the same substitutions had little effect on EBOV GPcl binding, consistent with the absence of loop 3 contacts in EBOV (Extended Data Fig. 7d and Supplementary Fig. 3).

We also examined other NPC1-C residues that differentially interact with the two GPs. His418 and Ile419 of NPC1-C formed favourable interactions with the RAVV RBS (Extended Data Fig. 8c,d), and alanine substitutions at these sites reduced RAVV GPcl binding (Extended Data Fig. 7d). By contrast, Lys498 and Asp508 of NPC1-C engage in an unfavourable intramolecular interaction (Extended Data Fig. 8e); alanine substitutions at these sites increased RAVV GPcl binding (Extended Data Fig. 7d). At the EBOV interface, these residues behaved differently: His418 made no contact, Ile419 and Lys498 each contributed favourably, and Asp508 contributed unfavourably (Extended Data Fig. 8c−e). Correspondingly, alanine substitutions had no effect, reduced affinity,

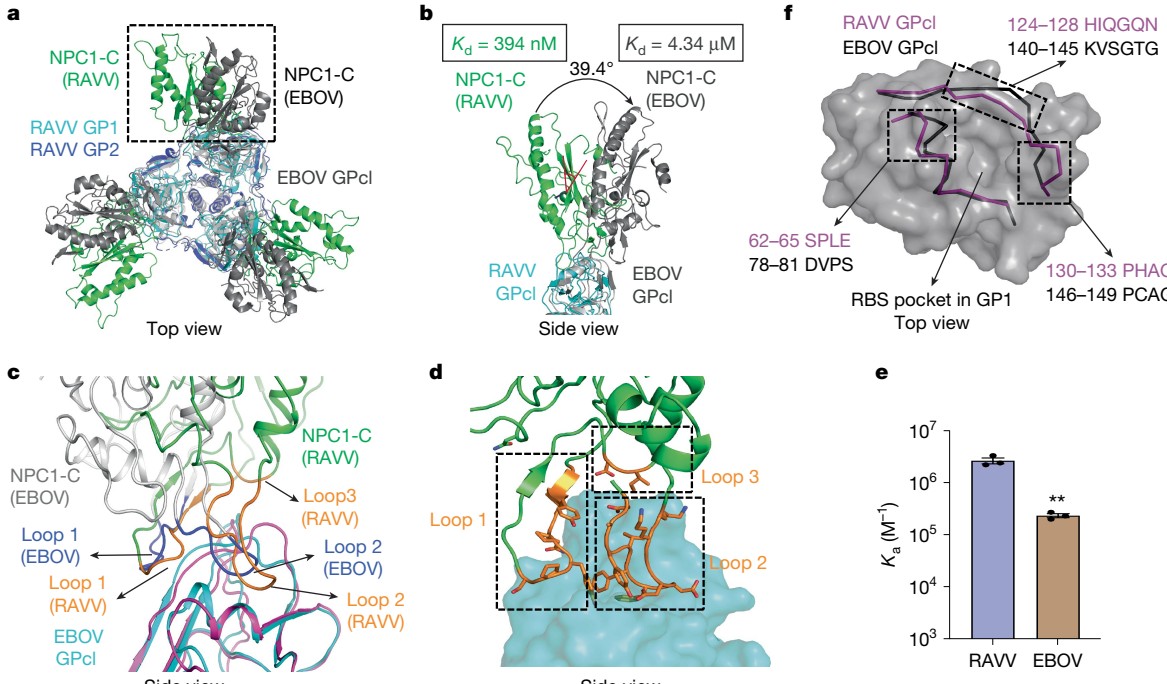

**Fig. 3 | Structural basis for NPC1 binding by MBV GP. a**, Structural alignment of NPC1-C-bound RAVV GPcl and NPC1-C-bound EBOV GPcl (PDB ID: 5F1B). The two structures are aligned based on their GPcl regions. GP1 and GP2 of RAVV GPcl are shown in cyan and blue, respectively, whereas NPC1-C bound to RAVV GPcl is shown in green. EBOV GPcl and its associated NPC1-C are shown in grey. **b**, Enlarged view of NPC1-C bound to RAVV GPcl and EBOV GPcl, as in **a**, highlighting differences in their binding orientations. $K_d$ values for the two GPcls binding to NPC1-C are shown. **c**, Structural alignment of NPC1-C-bound RAVV GPcl and NPC1-C-bound EBOV GPcl. RAVV GPcl is coloured in magenta, with bound NPC1-C in green and its three RBS-binding loops in orange. EBOV GPcl is coloured cyan, with bound NPC1-C in grey and its two RBS-binding loops in blue. **d**, NPC1-C interacts with RAVV GPcl through three loops. The RAVV RBS is shown as a cyan surface representation, NPC1-C as a green ribbon and the three RBS-binding loops as orange ribbons. **e**, Binding affinities ($K_a$) of RAVV and EBOV GPcls for NPC1-C. $K_a$ values were measured by SPR in three independent replicates (see Supplementary Figs. 2 and 3 for full curves). Data are presented as mean ± s.e.m. ($n$ = 3 independent experiments). Statistical significance between RAVV and EBOV GPcls was determined using a two-tailed, unpaired Student's $t$-test, with the $P$ value as follows: RAVV versus EBOV $P$ = 0.0021. The asterisks above the bar indicate statistical significance (**$P$ < 0.01). **f**, Comparison of the RBS pocket opening in RAVV and EBOV GPcls. Relative to EBOV GPcl, three regions surrounding the RAVV RBS pocket undergo marked conformational changes. The residues in each region are listed.

reduced affinity and increased affinity, respectively (Extended Data Fig. 7d). Together, these differences — along with the new loop 3 contacts — explain why RAVV GPcl binds to NPC1-C with substantially higher affinity than EBOV GPcl.

To further understand the distinct mode of NPC1 binding by RAVV GPcl, we compared the shapes of the RBS pocket openings in the RAVV and EBOV GPcls (Fig. 3f). Relative to EBOV GPcl, three regions surrounding the RAVV RBS pocket undergo pronounced conformational changes. First, residues 130–133 swing outwards, a rearrangement probably facilitated by the loss of the EBOV disulfide bond (Fig. 3f). Moreover, previous computational analysis suggested that mutation of EBOV GPcl Gly149 to a bulkier residue would open the RBS (pushing the opening outwards) and indirectly create new interactions with NPC1. Our structure confirms this prediction: the corresponding Gln133 in MBV GP promotes outwards movement of the Pro130–Gln133 loop towards NPC1-C loop 3, forming new contacts that are absent in EBOV GPcl (Extended Data Fig. 8b). The other two changes involve residues 62–65 and 124–128; extensive residue substitutions in these segments drive them outwards and inwards, respectively (Fig. 3f). Collectively, these reshaped RBS pocket openings in RAVV GPcl create additional NPC1 contacts and compel NPC1-C to bind in a distinct, higher-affinity orientation.

In addition to these amino acid differences, RAVV GP contains an N-linked glycan at residue 94 (N94 glycan; Extended Data Fig. 5c), which is absent in EBOV GP. The N94 glycan of RAVV GP interacts with the N557 glycan of NPC1-C. However, mutating either glycan caused aggregation of recombinant RAVV GPcl or NPC1-C, preventing further functional analysis. As such, it remains unclear whether the N94 glycan

enhances or reduces NPC1-binding affinity, or whether it modulates GP interactions with cell-surface attachment factors (for example, lectins) during viral endocytosis.

## NPC1-triggered GP conformational changes

The structure of the RAVV GPcl–NPC1-C complex revealed substantial conformational changes in GP1 compared with the apo RAVV GPcl structure (Fig. 4a). NPC1-C binding induced outwards movements of several GP1 elements away from the trimeric centre (Fig. 4a,b). Consequently, in the NPC1-C-bound structure, the three protomers were positioned farther apart, with a reduced buried interface between them (decreasing from 1,587 to 1,402 Å²). This loosening of protomer packing probably facilitates the transition to the post-fusion state, thereby promoting membrane fusion.

Filovirus GPs belong to the class I family of membrane fusion proteins, whose pre-fusion trimers collapse into a post-fusion six-helix bundle. This mechanism differs from that of class II proteins (for example, dengue virus E), which transition from pre-fusion dimers to elongated β-sheet trimers, and class III proteins (for example, herpesvirus gB), which transition from pre-fusion trimers to post-fusion trimers with mixed class I–II features[26]. Because stabilizing mutations were introduced into the GP2 subunit of recombinant RAVV GPcl, it cannot complete the transition to the post-fusion six-helix bundle. Nevertheless, the high efficiency of pseudovirus entry mediated by RAVV GP — whether full-length GP (Fig. 1d), GP-ΔM or GPcl (Extended Data Fig. 9) — indicates that full structural transitions of RAVV GP occur on the pseudovirus surface during cell entry

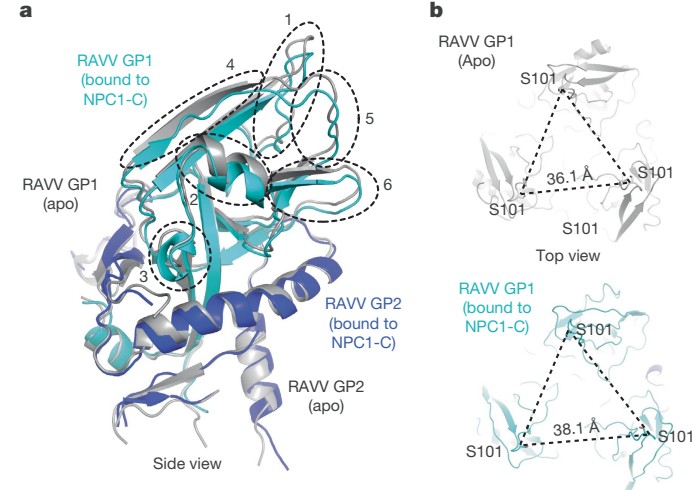

**Fig. 4 | Conformational changes in MBV GPcl triggered by NPC1 binding.**
**a**, Structural alignment of RAVV GPcl alone and RAVV GPcl bound to NPC1-C. GP1 and GP2 of NPC1-C-bound RAVV GPcl are shown in cyan and blue, respectively, whereas RAVV GPcl alone is coloured grey. Structural elements in NPC1-C-bound GPcl that undergo conformational shifts relative to GPcl alone are highlighted. **b**, Comparison of the distances between GP protomers in NPC1-C-bound RAVV GPcl (bottom) and RAVV GPcl alone (top), measured as the distances between the Ser101 residues in each GP1 protomer.

following NPC1 engagement in endosomes. The multiple roles of NPC1 in RAVV entry, including its high-affinity binding to GPcl and its ability to trigger conformational change, underscore the GPcl–NPC1 interface as a key target for antiviral inhibitors acting on either GPcl or NPC1.

## Nanobody neutralization of MBV entry

The structure of the RAVV GP-ΔM–Nanosota-MB1 complex revealed that three Nanosota-MB1 molecules bind to the trimeric GP-ΔM, with each nanobody engaging one RBS (Fig. 1c). Nanobodies contain three complementarity-determining regions (CDRs) and four framework regions. Nanosota-MB1 primarily uses CDR2 and CDR3 to engage the RBS (Fig. 5a and Extended Data Fig. 5d). CDR2 inserts into the hydrophobic cavity of the RBS, with Phe58 and Ile59 forming strong hydrophobic interactions with Trp70, Phe72 and Met154, and Ile95, Val97 and Ile125 of the RBS, respectively (Fig. 5a,b). This binding mode mimics NPC1-C, which inserts its loop 2 into the same cavity, using Phe503 and Phe504 to engage the same hydrophobic residues. Hence, Nanosota-MB1 achieves high-affinity RBS binding by mimicking NPC1. To validate these structural findings, we performed two biochemical assays. First, a competition SPR assay with RAVV GPcl, NPC1-C and Nanosota-MB1 showed that Nanosota-MB1 blocked NPC1-C binding to RAVV GPcl (Fig. 5c), confirming that they compete for the same RBS. Second, MBV pseudovirus neutralization assays demonstrated that Fc-tagged Nanosota-MB1 effectively neutralized pseudoviruses from RAVV and two MARV strains (Musoke and Angola; Fig. 5d). Together, these biochemical data corroborate our structural findings and establish that Nanosota-MB1 neutralizes MBV entry by directly blocking receptor binding.

## Discussion

Compared with EBOV, MBVs cause higher fatality rates in humans. MBV GP mediates viral entry into host cells, making it a key determinant of infectivity and pathogenesis. Here we have shown that MBV GP drives viral entry into human cells more efficiently than EBOV GP. This enhanced entry probably contributes to the high fatality rates of MBV, although additional factors are also involved. To investigate the structural basis of this enhanced entry, we determined cryo-EM

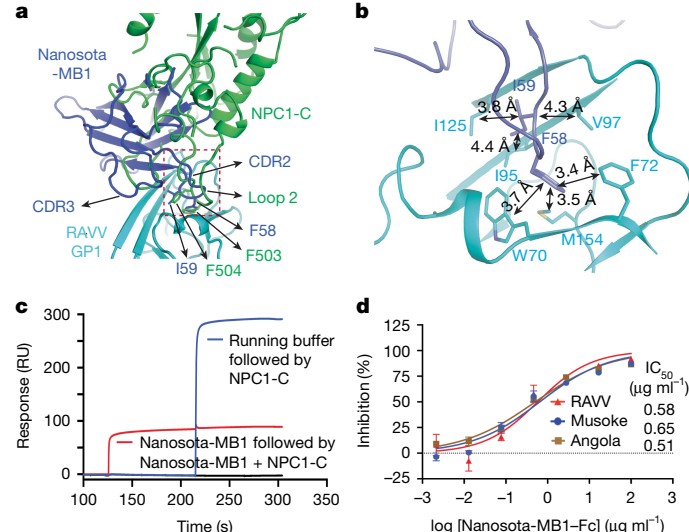

**Fig. 5 | Structural basis for nanobody neutralization of MBV entry.**
**a**, Structural alignment of RAVV GP1 in complex with the neutralizing nanobody Nanosota-MB1 and in complex with NPC1-C. RAVV GP1 is shown in cyan, Nanosota-MB1 in blue and NPC1-C in green. CDR2 and CDR3 of Nanosota-MB1 that directly engage the NPC1-C binding site are highlighted. Within CDR2, residues Phe58 and Ile59 insert into a hydrophobic pocket of the RAVV RBS. Loop 2 of NPC1-C, containing Phe504, inserts into the same hydrophobic pocket, illustrating the mimicry of NPC1 recognition by the nanobody. **b**, Enlarged view of the hydrophobic interactions (indicated by double arrows with distances labelled) formed between CDR2 of Nanosota-MB1 and the RAVV RBS pocket. **c**, Competition SPR analysis of NPC1-C and Nanosota-MB1 binding to RAVV GPcl. RAVV GPcl was immobilized on the sensor chip. The red curve shows the SPR signal when Nanosota-MB1 was injected first, followed by a mixture of Nanosota-MB1 and NPC1-C. The blue curve shows the SPR signal when buffer was injected first, followed by NPC1-C. **d**, Neutralization efficacy of Fc-tagged Nanosota-MB1 to MBV pseudoviruses. Retroviruses pseudotyped with full-length MBV GPs were used to infect Huh7 cells in the presence of serial dilutions of Fc-tagged Nanosota-MB1. Neutralization potency is expressed as the concentration required to inhibit pseudovirus entry by 50% ($IC_{50}$). Data are presented as mean ± s.e.m. ($n$ = 3 biologically independent samples).

structures of the RAVV GP ectodomain in three states: (1) GPcl (lacking the glycan cap and MLD), (2) GPcl bound to NPC1, and (3) GP-ΔM (lacking the MLD but retaining the glycan cap) in complex with the nanobody Nanosota-MB1, each representing one of the first structures of its kind. These structures reveal three critical functions of RAVV GP: glycan cap shielding, NPC1 recognition and NPC1-triggered conformational changes that promote membrane fusion.

Unexpectedly, our apo RAVV GPcl structure revealed a glycan cap loop occupying the RBS, despite complete proteolytic removal of the glycan cap. Previous studies have suggested that the glycan cap does not obstruct neutralizing antibody binding to the RBS, implying full flexibility[13,19]. However, this interpretation contradicts the presumed role of the glycan cap in shielding the RBS from immune recognition. Our findings indicate that the flexibility of the glycan cap is partially constrained by interactions with the RBS. Functionally, the RAVV glycan cap fully blocks NPC1 binding but only partially obstructs nanobody binding. By contrast, the EBOV glycan cap completely blocks access to both NPC1 and neutralizing antibodies or nanobodies. This raises an evolutionary question regarding what advantage RAVV gains from a glycan cap with partial flexibility, allowing only limited immune evasion. One possibility is that increased flexibility facilitates glycan cap removal in endosomes, thereby enhancing NPC1 binding and viral entry. Indeed, previous work − including ours − has shown that stabilizing the glycan cap impairs proteolysis, reduces RBS exposure and inhibits EBOV GP-mediated entry[25,27]. If the same applies to MBV GP, this would

suggest that RAVV has evolved a partially flexible glycan cap to balance immune evasion with efficient entry.

Our structure of RAVV GPcl in complex with human NPC1 revealed striking differences in receptor engagement compared with EBOV GPcl. First, NPC1 binds to the RAVV RBS at a distinct angle. Second, whereas NPC1-C engages the EBOV RBS with two protruding loops (loops 1 and 2), it engages the RAVV RBS with three loops, including an additional loop 3 that provides extra anchoring and alters the binding orientation. Third, sequence divergence between the RAVV and EBOV RBSs generates substantially different interaction networks and reshapes the openings of the RBS pocket. Finally, these differences enable RAVV GPcl to bind to NPC1-C with approximately 11-fold higher affinity than EBOV GPcl. Guided by these structural insights, we performed mutagenesis studies that confirmed the important roles of NPC1 residues differentially interacting with RAVV and EBOV GPcls. Collectively, our data demonstrate that RAVV GPcl uses distinct structural mechanisms to engage NPC1, resulting in substantially stronger receptor binding than EBOV GPcl. We note that although a positive relationship between receptor affinity and entry efficiency has been clearly established for some other virus families, such as coronaviruses[28,29], this relationship is less well defined for filoviruses because of the complexity of the filovirus entry pathway.

Our structural analysis revealed substantial NPC1-induced conformational changes in RAVV GPcl. Compared with the apo structure, NPC1 binding triggers substantial GP1 rearrangements, increasing protomer separation and reducing inter-protomer packing. This shift probably lowers the energy barrier for the pre-fusion-to-post-fusion transition, thereby facilitating membrane fusion. NPC1-induced conformational changes in EBOV GPcl remain unclear due to the lack of appropriate structures for direct comparison. However, these changes may be more pronounced in RAVV GPcl due to its potentially lower structural rigidity, attributed to the absence of the Cys121–Cys147 bond found in EBOV GPcl, as well as its distinct NPC1-binding mode and enhanced NPC1-binding affinity compared with EBOV GPcl. The pronounced susceptibility of RAVV GPcl to NPC1-induced structural transitions may further enhance its membrane fusion capability.

In addition, we identified the nanobody targeting MBVs: Nanosota-MB1. Our structure of the RAVV GP-ΔM–Nanosota-MB1 complex reveals molecular mimicry, with the nanobody binding the RBS through mechanisms similar to NPC1-C. Specifically, its CDR2 loop inserts into the hydrophobic cavity of the RBS and engages nearly the same residues as NPC1 loop 2. Nanosota-MB1 binds to RAVV GPcl with ultra-high affinity, outcompetes NPC1-C for the same site, and its Fc-tagged version neutralizes RAVV and two major MARV strains (IC$_{50}$ of approximately 0.5 µg ml$^{-1}$), showing much greater potency than known human antibodies (IC$_{50}$ of 2–200 µg ml$^{-1}$)[30]. Moreover, due to the partial flexibility of the glycan cap, Nanosota-MB1 can bind to MBV GP with high affinity even before endocytosis, further enhancing its efficacy. Although used here primarily as a research tool to probe MBV entry, Nanosota-MB1 also shows potential as a neutralizing therapeutic to MBVs.

In summary, our study has demonstrated that MBV GP mediates viral entry more efficiently than EBOV GP and reveals several unique structural features, including a partially flexible glycan cap, a distinct NPC1 binding mode, increased NPC1-binding affinity and pronounced susceptibility to NPC1-triggered conformational changes. These features, acting in combination, probably contribute to enhanced entry of MBVs and may partly help to explain their higher fatality rates. Together, our findings provide critical insights into MBV entry mechanisms and highlight potential strategies for intervention, with important implications for global health and pandemic preparedness.

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

# Methods

## Cell lines and plasmids

HEK293T and Huh7 cells (American Type Culture Collection (ATCC)) were maintained in Dulbecco's modified eagle medium supplemented with 10% fetal bovine serum, 2 mM L-glutamine, 100 U ml$^{-1}$ penicillin and 100 μg ml$^{-1}$ streptomycin. HUVEC cells (ATCC) were cultured in vascular cell basal medium supplemented with the Endothelial Cell Growth Kit-BBE (ATCC), 100 U ml$^{-1}$ penicillin and 100 μg ml$^{-1}$ streptomycin. THP-1 cells (ATCC) were cultured in RPMI-1640 medium supplemented with 10% fetal bovine serum, 0.05 mM 2-mercaptoethanol (Gibco), 100 U ml$^{-1}$ penicillin and 100 μg ml$^{-1}$ streptomycin. To induce macrophage-like differentiation, THP-1 cells were treated with 30 nM phorbol 12-myristate 13-acetate (PMA; Sigma-Aldrich) for 24 h, followed by a 24-h incubation in PMA-free medium[31]. Expi293F cells (Thermo Fisher) were grown in Expi293 Expression Medium (Thermo Fisher). ss320 *Escherichia coli* (Lucigen) and TG1 *E. coli* (Lucigen) were cultured in 2YT medium. HEK293T, Huh7 and THP-1 cells were authenticated by the vendors using short tandem repeat (STR) profiling. Authentication documentation for Expi293F and HUVEC cells was not available on the vendors' websites. HEK293T cells were tested for mycoplasma contamination in our laboratory and by the vendor and were negative in both cases. Huh7, HUVEC, THP-1 and Expi293F cells were tested and confirmed negative for mycoplasma by the vendors. No commonly misidentified cell lines were used.

Genes encoding RAVV GP (GenBank: ACD13005.1), Musoke MARV GP (NCBI Reference Sequence: YP_001531156.1), Angola MARV GP (GenBank: APQ46224.1), EBOV GP (NCBI RefSeq protein: NP_066246.1) and human NPC1 (UniProt: O15118) were synthesized (GenScript). For full-length GP pseudovirus production, GP genes were cloned into the pcDNA3.1(+) vector with or without a C-terminal C9 tag (the tag-free version was defined as wild type), as previously described[32]. For GP-ΔM pseudovirus production, the RAVV GP-ΔM gene (residues 1–636, lacking residues 257–425 corresponding to the MLD) was cloned into pcDNA3.1(+) with a C-terminal C9 tag. For protein expression, the RAVV GP-ΔM gene (residues 1–636, lacking residues 257–425 and containing a K589I mutation to stabilize GP2) and the EBOV GP-ΔM gene (residues 1–632, lacking residues 313–463 corresponding to the MLD) were each fused to a C-terminal foldon trimerization motif and His6 tag, and cloned into the Lenti-CMV vector, as previously described[25,33]. The human NPC1-C gene (residues 374–620, wild type or containing introduced mutations) was fused to a C-terminal His6 tag, and the gene encoding Nanosota-MB1 was fused to a C-terminal human IgG1 Fc tag (GenBank: AEV43323.1); both were cloned into the Lenti-CMV vector, as previously described[25,32].

## Preparation of GP and NPC1

RAVV GP-ΔM and human NPC1-C proteins were expressed and purified as previously described[19,25,32]. Plasmids (500 μg) encoding GP-ΔM or NPC1-C were transiently transfected into 500 ml Expi293F cells using 1.5 ml polyethylenimine (Polysciences). Three days post-transfection, supernatants were collected, and proteins were purified on a Ni-NTA column (Cytiva) with an imidazole gradient in PBS. Further purification was performed by size-exclusion chromatography: GP-ΔM on a Superose 6 column (Cytiva) and NPC1-C on a Superdex 200 column (Cytiva), both in buffer containing 20 mM Tris (pH 7.4) and 200 mM NaCl. Purified proteins were flash-frozen in liquid nitrogen and stored at −80 °C. To generate RAVV GPcl, 3 mg of RAVV GP-ΔM was treated with 60 μg of trypsin (Sigma-Aldrich) for 1 h, followed by purification on a Superose 200 column (Cytiva). To generate EBOV GPcl, 3 mg of EBOV GP-ΔM was treated overnight with 15 μg of thermolysin L (Sigma-Aldrich)[25], followed by purification on a Superose 200 column (Cytiva).

## Preparation of nanobodies

RAVV GP-targeting nanobodies were generated as previously described[25,34,35]. An alpaca was immunized subcutaneously with RAVV GP-ΔM at 2-week intervals for a total of seven immunizations (Turkey Creek Biotechnology; animal protocol 18-03, in accordance with institutional and national guidelines for the care and use of research animals). Following immunization, blood was collected, and peripheral blood mononuclear cells were isolated (Vanderbilt Antibody and Protein Resource Core). A cDNA library was constructed from peripheral blood mononuclear cell RNA using oligo(dT) primers and Superscript IV reverse transcriptase (Thermo Fisher). Nanobody genes were amplified by nested PCR and cloned into a modified pADL22 vector (Antibody Design Labs). Ligation products were electroporated into *E. coli* TG1 to generate the nanobody phage display library, following the manufacturer's protocol (Antibody Design Labs).

Screening of the nanobody phage display library was performed as previously described[25,34,35]. Three rounds of bio-panning were carried out to enrich for nanobodies binding to RAVV GP-ΔM. For each round, 20 μg of purified GP-ΔM was coated onto an immune tube overnight. The tube was blocked with 5% milk, incubated with 500 μl of phages for 1 h, and washed; the retained phages were eluted and used to infect *E. coli* TG1. Amplified phages were used for subsequent rounds. After the third round, eluted phages were used to infect *E. coli* ss320, which were plated on 2YT agar. Single colonies were picked, and nanobody expression was induced with 1 mM IPTG. Supernatants were screened by ELISA to identify GP-ΔM-binding nanobodies.

ELISA was performed as previously described[25,34,35]. In brief, plates were coated with 100 ng of RAVV GP-ΔM overnight and blocked with 2% BSA. Plates were then sequentially incubated with supernatants from *E. coli* ss320 expressing haemagglutinin (HA)-tagged nanobodies and with horseradish peroxidase-conjugated anti-HA antibody (Sigma-Aldrich). ELISA substrate (Invitrogen) was added, and reactions were stopped with 1 N H$_2$SO$_4$. Absorbance at 450 nm (A$_{450}$) was measured using a Synergy LX Multi-Mode Reader (BioTek).

His-tagged nanobodies were expressed and purified from the periplasm of *E. coli* ss320 as previously described[25,34,35]. Expression was induced with 1 mM IPTG. Cell pellets were collected, resuspended in 15 ml TES buffer (0.2 M Tris, pH 8.0, 0.5 mM EDTA and 0.5 M sucrose) and shaken on ice for 1 h. The suspension was then diluted with 40 ml of one-quarter of TES buffer (each component at one-quarter concentration) and shaken on ice for another hour. Nanobodies in the supernatant were purified sequentially on a Ni-NTA column (Cytiva) followed by a Superdex 200 column (Cytiva).

Fc-tagged nanobodies were expressed and purified from the supernatant of Expi293F cells as previously described[25,34,35]. Plasmids were transiently transfected into Expi293F cells using polyethylenimine (Polysciences). Three days post-transfection, supernatants were harvested, and nanobodies were purified on a protein A column (Cytiva), followed by further purification on a Superdex 200 column (Cytiva).

## SPR

SPR was performed to measure binding affinities between RAVV GP (GP-ΔM or GPcl) and its ligands (NPC1-C or Nanosota-MB1), as well as between EBOV GPcl and NPC1-C, using a Biacore S200 system (Cytiva) as previously described[36,37]. Recombinant GP was immobilized on a CM5 sensor chip (Cytiva) via chemical crosslinking. NPC1-C (0.156–2.5 μM for RAVV GPcl; 0.625–10 μM for EBOV GPcl), NPC1-C mutants (at various concentrations) or Nanosota-MB1 (0.02–0.32 μM for RAVV GPcl and GP-ΔM) were injected in running buffer containing 50 mM MES (pH 6.0), 150 mM NaCl and 0.05% Tween-20. Binding responses were recorded as response units. Binding data were analysed using Biacore Evaluation Software (Cytiva).

SPR was also used to assess competition between NPC1-C and Nanosota-MB1 for RAVV GPcl binding, as previously described[35]. RAVV GPcl was immobilized on a CM5 sensor chip, and 1 μM Nanosota-MB1 was first injected to saturate the chip. This was followed by injection of a mixture containing 1 μM Nanosota-MB1 and 10 μM NPC1-C. For the

control, running buffer was injected first, followed by 10 μM NPC1-C. Competitive binding was evaluated by comparing the SPR signals from the Nanosota-MB1–NPC1-C mixture with those from NPC1-C alone.

## Pseudovirus entry assay

Pseudovirus entry assays were performed to evaluate entry efficiencies of MBV and EBOV pseudoviruses, as previously described[25]. Pseudoviruses bearing either wild-type or C9-tagged full-length GP were generated by co-transfecting HEK293T cells with a pcDNA3.1(+) plasmid encoding GP, the helper plasmid psPAX2 encoding the HIV backbone and the reporter plasmid plenti-CMV-luc. After 72 h, pseudoviruses were harvested and used to infect Huh7 cells, HUVECs and THP-1-derived macrophages.

For neutralization assays, Fc-tagged Nanosota-MB1 at varying concentrations was mixed with MBV pseudoviruses before infection of Huh7 cells. After 48 h, cells were lysed, transferred to new plates and incubated with luciferase substrate. Relative light units were measured using an EnSpire plate reader (PerkinElmer). The Fc-tag was included to enhance nanobody multivalency for GP interactions and to increase in vivo half-life, while preserving the single-domain structure for antigen binding. The resulting construct remains approximately half the size of IgGs, making it compatible with intranasal administration.

GP expression in pseudoviruses was evaluated by western blot using anti-C9 antibody (Santa Cruz Biotechnology) for C9-tagged GPs, alpaca serum against EBOV GP (wild type or C9 tagged) from our previous study[25], and alpaca serum against RAVV and Musoke GPs (wild type or C9 tagged) from the current study.

To investigate entry efficiencies of GP-ΔM and GPcl pseudoviruses, C9-tagged RAVV GP-ΔM was used to generate GP-ΔM pseudoviruses. GP-ΔM pseudoviruses were then treated with trypsin as described for recombinant GP-ΔM protein to generate GPcl pseudoviruses. Both GP-ΔM and GPcl pseudoviruses were subsequently used to infect Huh7 cells.

## Cryo-EM

RAVV GPcl (approximately 3.0 mg ml⁻¹), the RAVV GPcl–NPC1-C complex (approximately 2.0 mg ml⁻¹) and the RAVV GP-ΔM–Nanosota-MB1 complex (approximately 3.0 mg ml⁻¹) were used for cryo-EM analysis as previously described[38]. The cryo-EM buffer consisted of 50 mM MES (pH 6.0) and 150 mM NaCl, matching the buffer conditions used in the SPR experiments. Before grid preparation, 8 mM CHAPSO was added to the samples. A 4 μl aliquot of each sample was applied to freshly glow-discharged Quantifoil R1.2/1.3 300-mesh copper grids (Electron Microscopy Sciences), blotted for 4 s at 22 °C under 100% humidity, and plunge-frozen in liquid ethane using a Vitrobot Mark IV (FEI). Images were acquired at the Hormel Institute, University of Minnesota[39], using a K3 Summit detector (Gatan) in super-resolution mode with binning 2 and correlated double sampling, along with a Gatan BioContinuum GIF energy filter (slit width of 20 eV). Data collection was performed with EPU software (Thermo Fisher) at a pixel size of 0.664 Å (nominal magnification of ×130,000) and a nominal defocus range of −1.0 to −2.0 μm. Each image consisted of 40 dose-fractionated frames, recorded with a total electron dose of 50 e⁻ Å⁻². Cryo-EM data collection statistics are summarized in Extended Data Table 1.

Cryo-EM data were processed using cryoSPARC (v4.5.1)[40], following the workflow outlined in Extended Data Figs. 1–3. All movies were motion corrected using MotionCor2 (ref. 41), and contrast transfer function parameters were estimated with CTFFIND (v4.1.13)[42], with data downsampled to three-quarters resolution (0.885333 Å per pixel after downsampling). Images with defocus values outside the range of −0.6 to −3.2 μm or with contrast function transfer fits worse than 7 Å were excluded. Particles were initially selected using the Blob and Template pickers in cryoSPARC (v4.5.1), followed by three rounds of 2D classification to remove junk particles. Good 2D classes were used for ab initio reconstruction of four maps, followed by heterogeneous refinement. After two rounds of 3D classification, particles from high-quality classes were subjected to non-uniform and contrast function transfer refinement, yielding the final maps. Post-processing with CryoFEM[43] was performed for the RAVV GPcl–NPC1-C complex to further enhance map density. Map resolutions were determined using gold-standard Fourier shell correlation at 0.143 between the two half-maps. Local resolution estimates were calculated using cryoSPARC (v4.5.1).

Initial model building for the RAVV GPcl, RAVV GPcl–NPC1-C complex and RAVV GP-ΔM–Nanosota-MB1 complex was performed in Coot (v0.8.9)[44] using the structure with PDB ID 6BP2 as the starting model. The initial model of Nanosota-MB1 was generated using the Swiss Model online tool (https://swissmodel.expasy.org/). Refinement was performed using Phenix (v1.16)[45], with additional manual adjustments in Coot (v0.8.9). Model and map statistics are summarized in Extended Data Table 1. Figures were generated using UCSF ChimeraX (v0.93)[46] and the PyMOL Molecular Graphics System (v3.0; Schrödinger)[47]. The buried interfaces between GP protomers were analysed using PDBePISA (https://www.ebi.ac.uk/pdbe/pisa/). Contact residues at protein–protein interfaces were identified using LigPlot[48].

## Reporting summary

Further information on research design is available in the Nature Portfolio Reporting Summary linked to this article.

## Data availability

The atomic models and corresponding cryo-EM density maps have been deposited to the PDB and the Electron Microscopy Data Bank, respectively, with accession numbers 9NPS and EMD-49631 (RAVV GPcl), 9NPR and EMD-49630 (RAVV GPcl–NPC1-C), and 9NPT and EMD-49632 (RAVV GP-ΔM–Nanosota-MB1). Source data are provided with this paper.

## Code availability

No custom code was used in this study.

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

**Acknowledgements** This study was supported by the US National Institutes of Health grant U19AI171954 (to F.L., B.L. and L.D.).

**Author contributions** G. Ye and F.B. conceptualized the project, expressed and purified the proteins, performed the cryo-EM, carried out the functional assays and reviewed the manuscript. H.T.-H. and M.H. expressed and purified the proteins, carried out the functional assays and reviewed the manuscript. L.D. conceptualized the project and reviewed the manuscript. G. Yang performed the cryo-EM and reviewed the manuscript. B.L. conceptualized the project, performed the cryo-EM and reviewed the manuscript. F.L. conceptualized and supervised the project, provided resources, guided the experiments and data analysis, and wrote the manuscript.

**Competing interests** The University of Minnesota has filed a US provisional patent application (application no. 63/775,149) covering Nanosota-MB1, with F.L., G. Ye. and F.B. as inventors. The other authors declare no competing interests.

**Additional information**
**Correspondence and requests for materials** should be addressed to Gang Ye, Bin Liu or Fang Li.

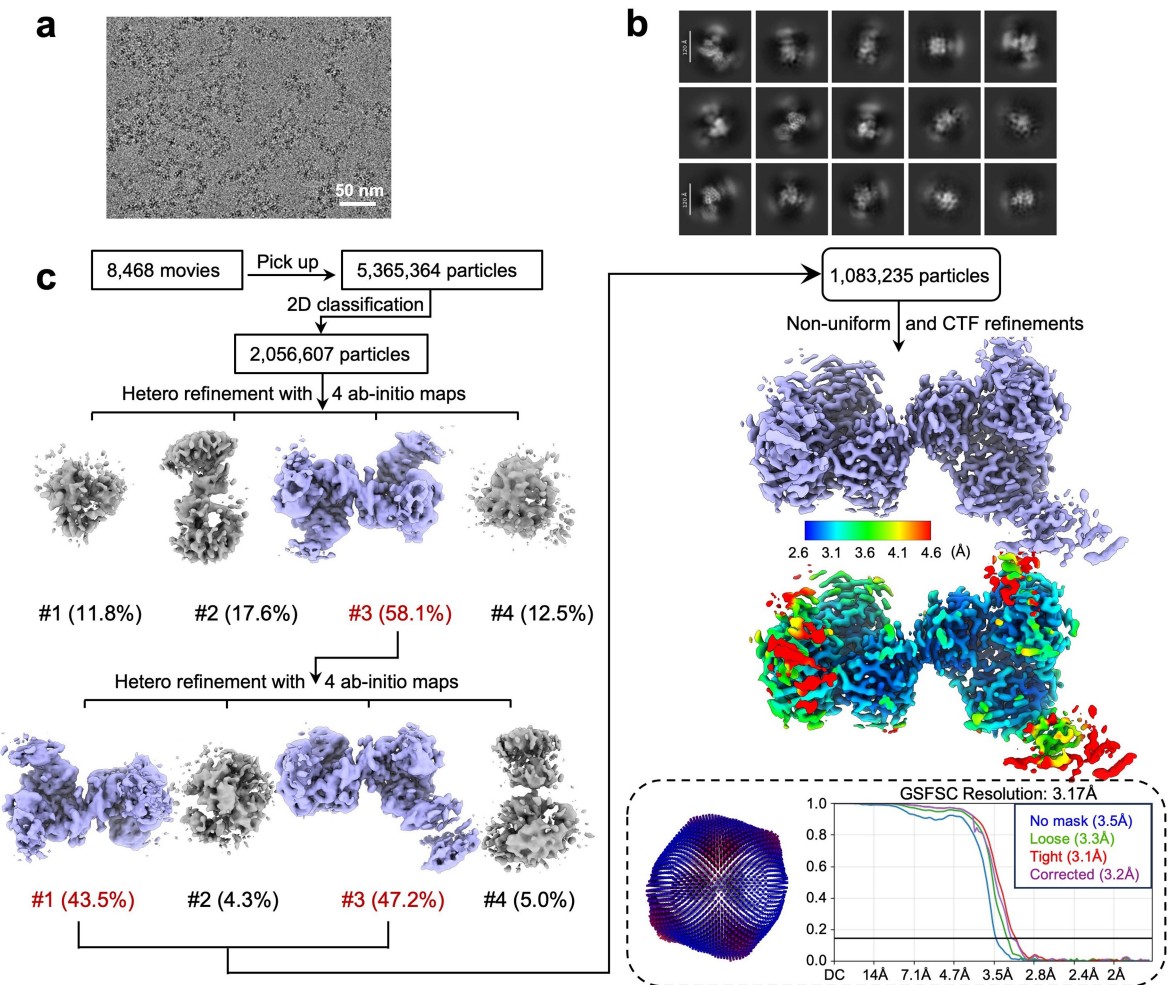

**Extended Data Fig. 1 | Workflow for cryo-EM image processing and map reconstruction of RAVV GPcl.** (**a**) Representative raw cryo-EM images and (**b**) 2D class averages of the protein are shown. (**c**) 3D refinement using all particles from high-quality 3D classes yielded a 3.17 Å resolution map. The final maps, half-map FSC curves, angular distribution plot, and corresponding local resolution illustrations are enclosed within the dashed black box. Representative cryo-EM micrographs were selected from a dataset comprising 8,468 independently acquired movies, all of which showed similar particle distributions and yielded consistent reconstructions.

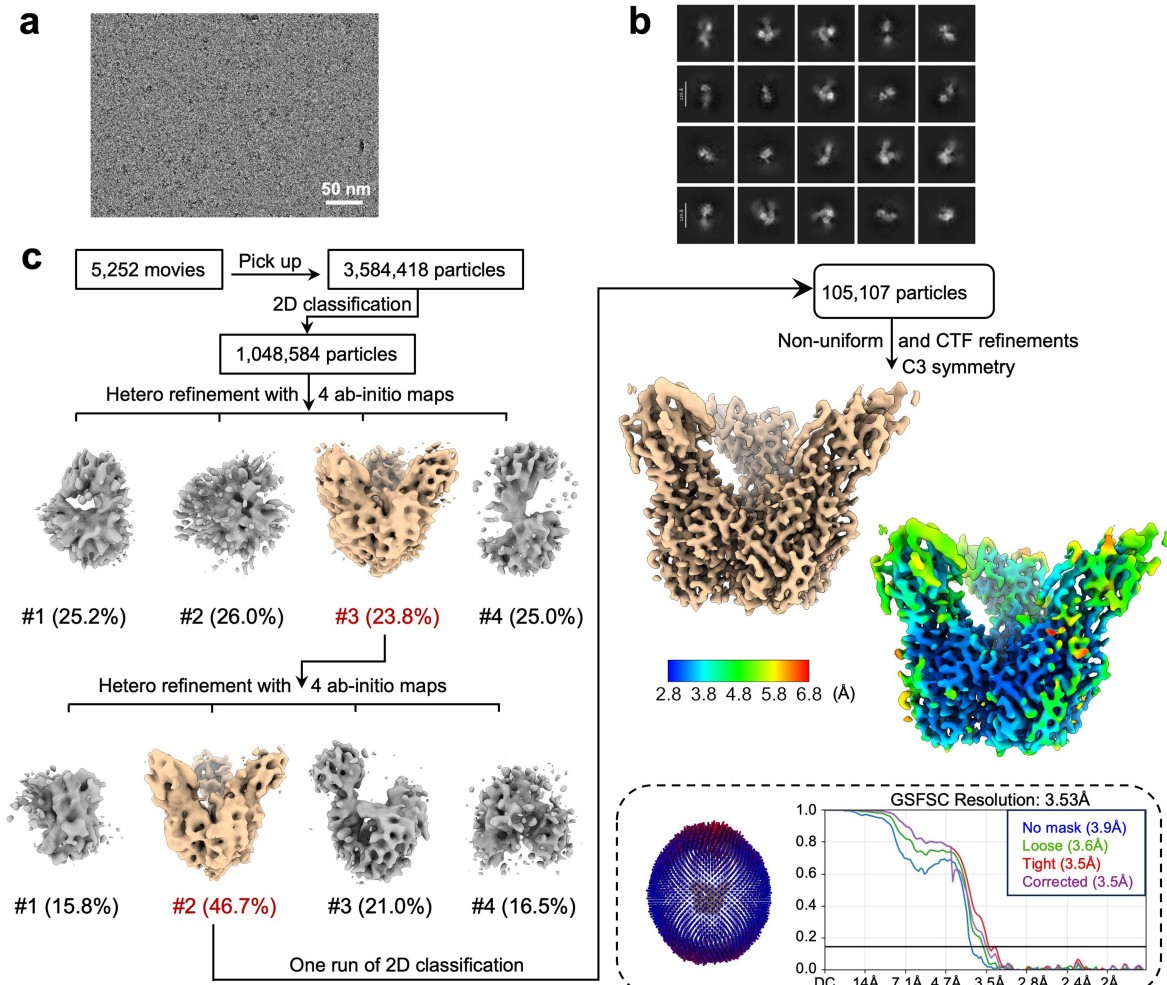

**Extended Data Fig. 2 | Workflow for cryo-EM image processing and map reconstruction of the RAVV GPcl/NPC1-C complex.** (**a**) Representative raw cryo-EM images and (**b**) 2D class averages of the complex are shown. (**c**) 3D refinement using particles from high-quality 3D classes yielded a 3.53 Å resolution map. The final maps, half-map FSC curves, angular distribution plot, and corresponding local resolution illustrations are enclosed within the dashed black box. Representative cryo-EM micrographs were selected from a dataset comprising 5,252 independently acquired movies, all of which showed similar particle distributions and yielded consistent reconstructions.

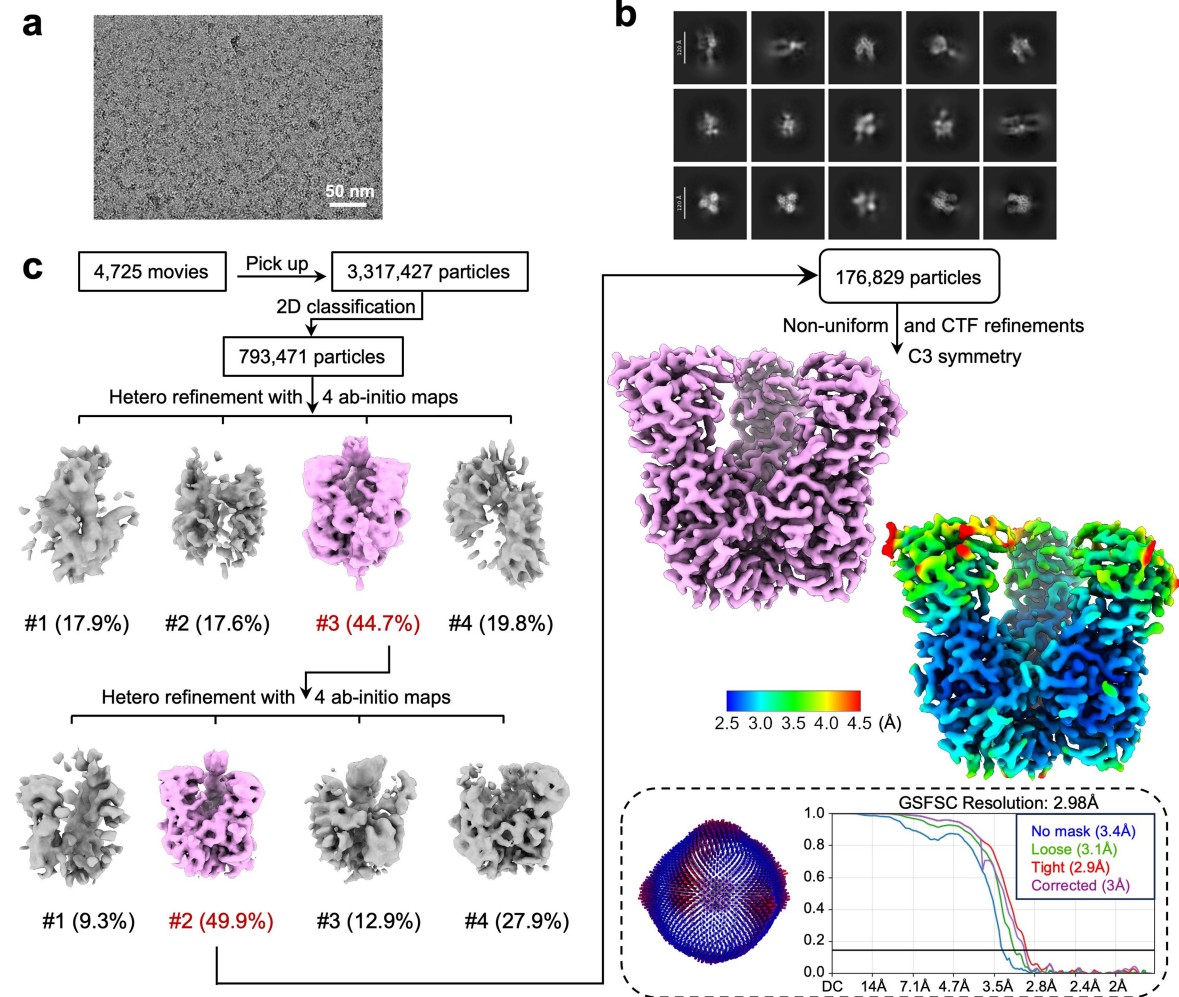

**Extended Data Fig. 3 | Workflow for cryo-EM image processing and map reconstruction of the RAVV GP-ΔM/Nanosota-MB1 complex.**
(**a**) Representative raw cryo-EM images and (**b**) 2D class averages of the complex are shown. (**c**) 3D refinement using particles from high-quality 3D classes yielded a 2.98 Å resolution map. The final maps, half-map FSC curves, angular distribution plot, and corresponding local resolution illustrations are enclosed within the dashed black box. Representative cryo-EM micrographs were selected from a dataset comprising 4,725 independently acquired movies, all of which showed similar particle distributions and yielded consistent reconstructions.

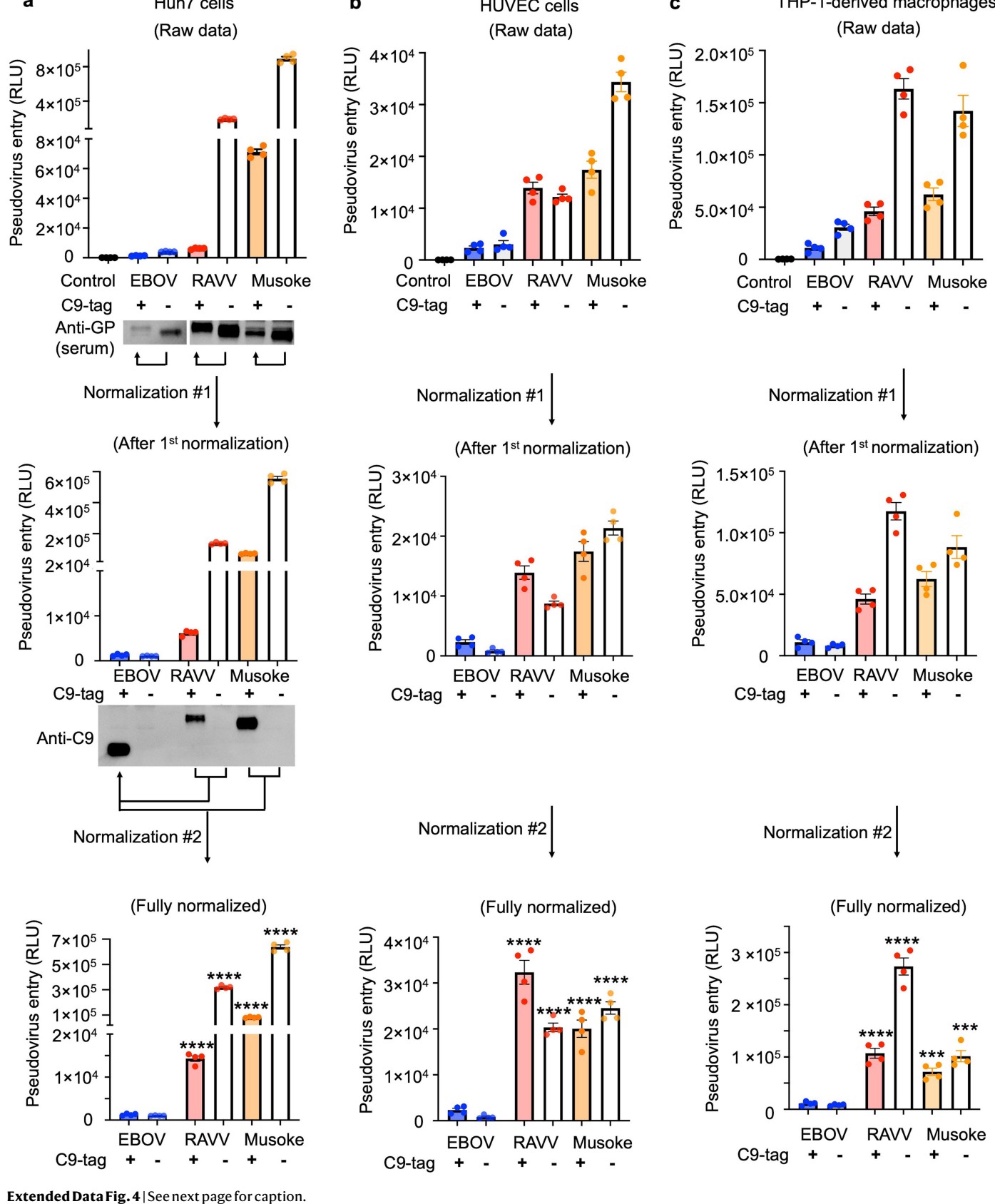

**Extended Data Fig. 4** | See next page for caption.

**Extended Data Fig. 4 | Entry efficiencies of filovirus pseudoviruses into three human target cell types, measured using a double-normalization strategy to control for GP expression levels.** Retroviruses pseudotyped with EBOV, RAVV, or Musoke GPs were used to infect Huh7 (hepatoma) (**a**), HUVEC (primary vascular endothelial) (**b**), and THP-1–derived macrophages (**c**), all major cellular targets of filoviruses. For each GP, both wild-type and C-terminally C9-tagged versions were packaged, yielding six pseudoviruses in total. Entry efficiencies were measured in four biological replicates per pseudovirus and cell type. Normalization was performed in two steps: first, wild-type pseudoviruses were standardized to their C9-tagged counterparts using GP-specific neutralizing sera; second, C9-tagged RAVV and Musoke pseudoviruses were normalized to C9-tagged EBOV pseudoviruses using anti-C9 antibodies. This double-normalization ensured equivalent GP expression levels across all pseudoviruses, allowing direct comparison of entry efficiencies mediated by the three wild-type GPs. Data are presented as mean ± SEM (n = 4 biologically independent samples). Statistical differences between wild-type EBOV GP and wild-type MBV GPs (RAVV and Musoke), and between C9-tagged EBOV GP and C9-tagged MBV GPs (RAVV and Musoke), were determined using two-tailed unpaired Student's t-tests. No adjustment was made for multiple comparisons. The P values are as follows: wild-type GP–Huh7, EBOV vs RAVV P < 0.0001, EBOV vs Musoke P < 0.0001; HUVEC, EBOV vs RAVV P < 0.0001, EBOV vs Musoke P < 0.0001; THP-1-derived macrophages, EBOV vs RAVV P < 0.0001, EBOV vs Musoke P = 0.0001; C9-tagged GP–Huh7, EBOV vs RAVV P < 0.0001, EBOV vs Musoke P < 0.0001; HUVEC, EBOV vs RAVV P < 0.0001, EBOV vs Musoke P < 0.0001; THP-1-derived macrophages, EBOV vs RAVV P < 0.0001, EBOV vs Musoke P = 0.0002. Asterisks above the bars indicate statistical significance (****P < 0.0001, ***P < 0.001). Uncropped Western blot images with molecular weight markers are shown in Supplementary Fig. 1. All experiments were repeated three times with independently prepared pseudovirus stocks, yielding consistent results.

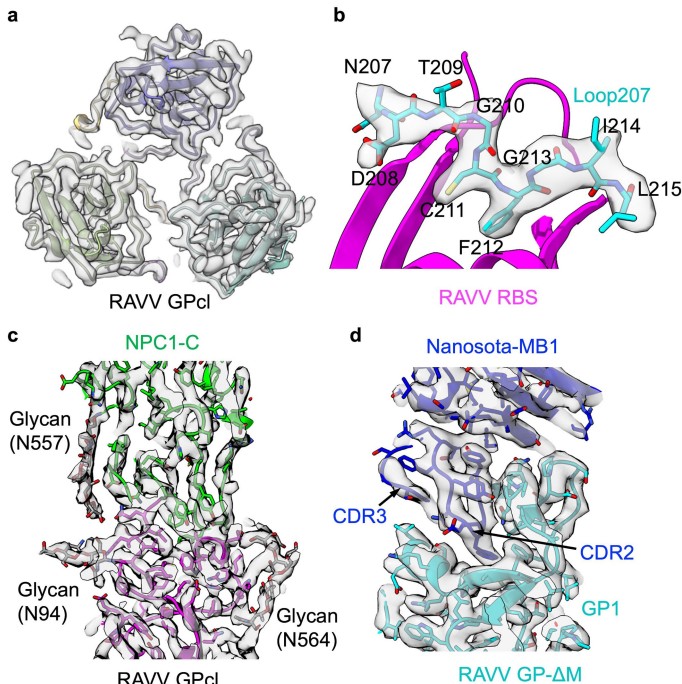

**a**

RAVV GPcl

**b**

N207  T209
G210
Loop207
D208
I214
G213
C211
L215
F212
RAVV RBS

**c**

NPC1-C
Glycan (N557)
Glycan (N94)
Glycan (N564)
RAVV GPcl

**d**

Nanosota-MB1
CDR3
CDR2
GP1
RAVV GP-ΔM

**Extended Data Fig. 5 | Cryo-EM densities of RAVV GP. (a)** Cryo-EM density map of trimeric RAVV GPcl (top view). **(b)** Cryo-EM density of the glycan cap loop in RAVV GPcl bound to the RAVV RBS (side view). **(c)** Cryo-EM density of the RAVV GPcl/NPC1-C complex (side view), highlighting three glycans: N94 and N564 from RAVV GPcl, and N557 from NPC1-C. **(d)** Cryo-EM density of the RAVV GP-ΔM/Nanosota-MB1 complex (side view), with CDR2 and CDR3 of Nanosota-MB1 highlighted.

**a**

```
                RBS1                              RBS2
RAVV    61  DSPLEASKRW AFRTGVPPKN    91  TCYNISVTDP SGKSL
Musoke  61  DSPLEASKRW AFRTGVPPKN    91  TCYNISVTDP SGKSL
Angola  61  DSPLEASKRW AFRAGVPPKN    91  TCYNISVTDP SGKSL
EBOV    77  TDVPSATKRW GFRSGVPPKV   107  NCYNLEIKKP DGSEC
             .  .*:*** .**:*****        .***:.:..* .*..

                        RBS3
RAVV   121  TVHHIQGQNP HAQGIALHLW   151  STTMYRGKVF
Musoke 121  TIHHIQGQNP HAQGIALHLW   151  STTMYRGKVF
Angola 121  TIHHIQGQNP HAQGIALHLW   151  STTMYRGKVF
EBOV   137  YVHKVSGTGP CAGDFAFHKE   167  STVIYRGTTF
             :*::.* .*  *  .:*:*        **.:***..*
```

**b**

| | NPC1-C Loop1 residues | | | | | | | |
|---|---|---|---|---|---|---|---|---|
| Binding to RAVV | H418 | I419 | Y420 | Q421 | P422 | Y423 | | S425 | |
| Binding to EBOV | | | Y420 | Q421 | P422 | Y423 | P424 | S425 | G426 |

| | NPC1-C Loop2 residues | | | | | | | |
|---|---|---|---|---|---|---|---|---|
| Binding to RAVV | K498 | D501 | D502 | F503 | F504 | V505 | Y506 | D508 |
| Binding to EBOV | | D501 | D502 | F503 | F504 | V505 | Y506 | |

| | NPC1-C Loop3 residues | |
|---|---|---|
| Binding to RAVV | D436 | I437 |
| Binding to EBOV | | |

**Extended Data Fig. 6 | Contact residues involved in filovirus GP binding to human NPC1.** (**a**) Sequence alignment of the RBS in RAVV, two MARV strains (Musoke and Angola), and EBOV. Asterisks indicate fully conserved residues, colons indicate strongly conserved residues, and periods indicate weakly conserved residues. Receptor-binding residues are highlighted in magenta and grouped into three regions: RBS1, RBS2, and RBS3. These residues are fully conserved within the MBV genus (RAVV and MARV). Receptor-binding residues in EBOV RBS that differ from those in MBV RBS are highlighted in red. (**b**) List of NPC1-C residues that directly contact the RBS of RAVV or EBOV. These residues are located on three loops that interact with RAVV RBS and two loops that interact with EBOV RBS.

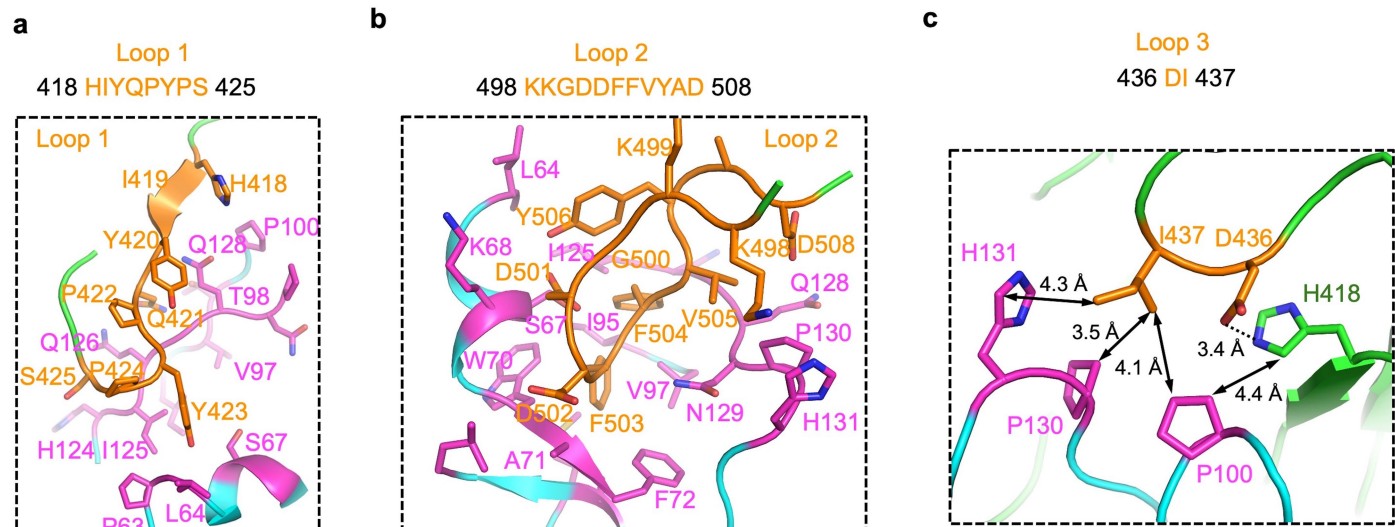

**a** Loop 1
418 HIYQPYPS 425

**b** Loop 2
498 KKGDDFFVYAD 508

**c** Loop 3
436 DI 437

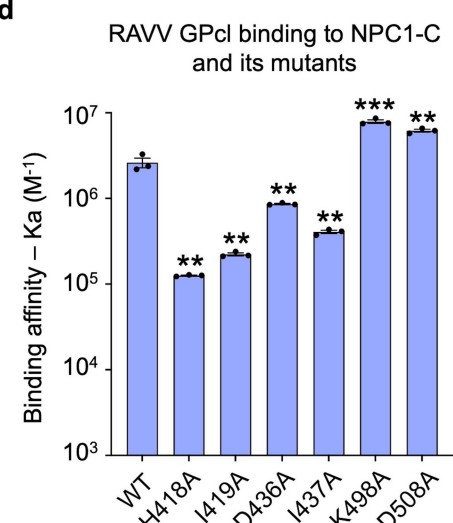

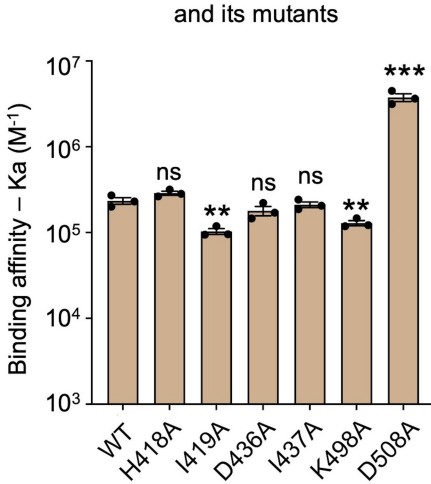

**Extended Data Fig. 7 | Detailed interactions between the three NPC1-C Loops and the RAVV RBS.** (**a**)–(**c**) The three RBS-binding loops are shown in orange. RAVV GPcl is colored cyan, with RBS residues that directly interact with NPC1-C highlighted in magenta. (**d**) Binding affinities ($K_A$) of RAVV and EBOV GPcls for NPC1-C and its mutants. NPC1-C residues that directly contact the RAVV RBS but not the EBOV RBS were individually mutated, and the resulting NPC1-C variants were analyzed by SPR to measure their binding affinities ($K_A$) for RAVV and EBOV GPcls (see Supplementary Figs. 2 and 3 for full SPR curves). Each SPR experiment was performed three times. Data are presented as mean ± SEM

(n = 3 independent experiments). Statistical significance between wild-type (WT) NPC1-C and each mutant was determined using two-tailed unpaired Student's t-tests, with P values as follows: for RAVV, WT vs H418A, P = 0.0018; WT vs I419A, P = 0.0021; WT vs D436A, P = 0.0065; WT vs I437A, P = 0.0028; WT vs K498A, P = 0.0005; WT vs D508A, P = 0.0010; for EBOV, WT vs H418A, P = 0.1063; WT vs I419A, P = 0.0041; WT vs D436A, P = 0.1450; WT vs I437A, P = 0.4384; WT vs K498A, P = 0.0095; WT vs D508A, P = 0.0008. Asterisks above the bars indicate statistical significance (***P < 0.001, **P < 0.01, ns = not significant).

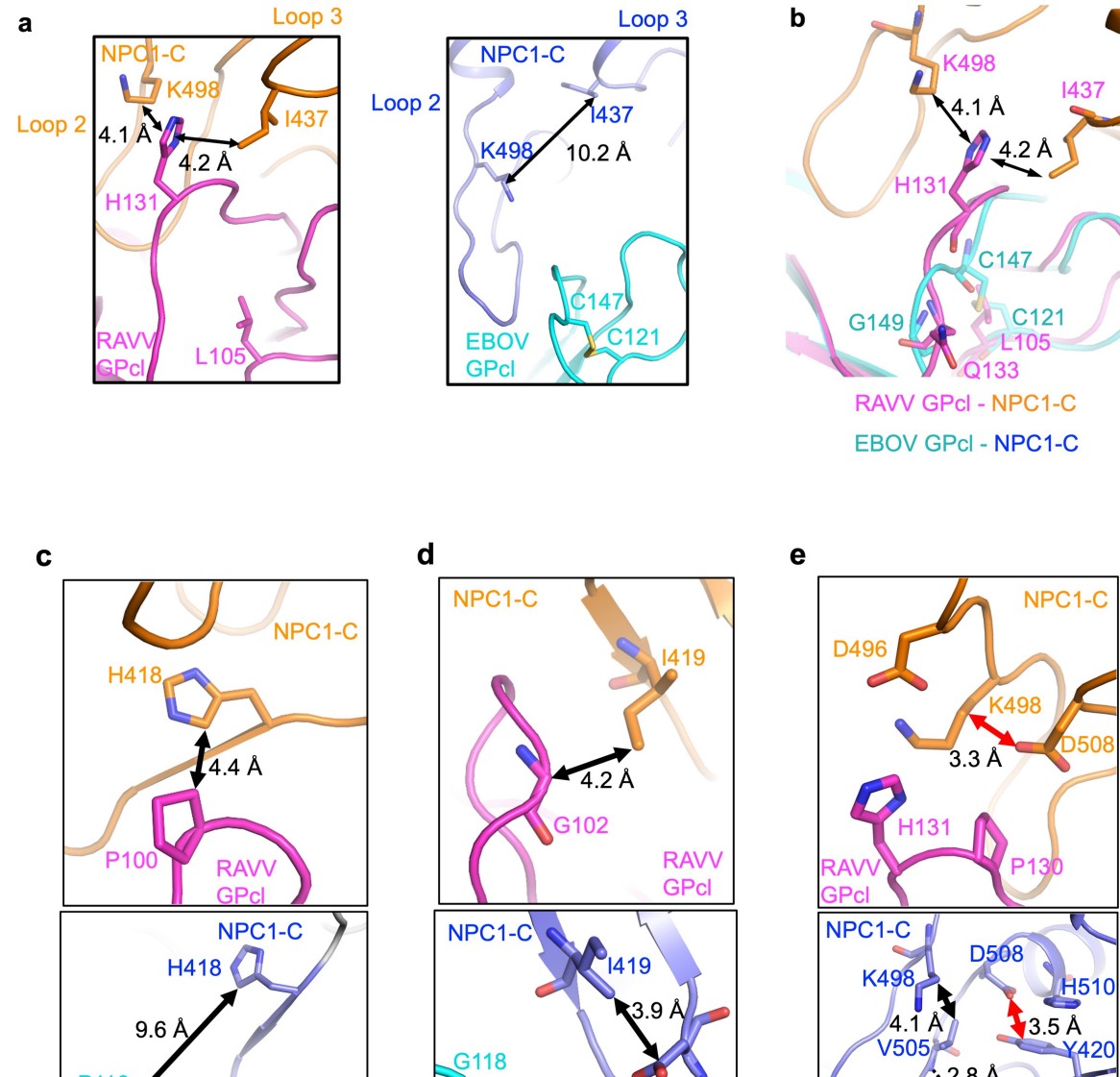

**Extended Data Fig. 8 | Comparison of differences in NPC1-C/RBS interaction networks between EBOV and RAVV GPcls.** (**a**) Structural changes in the RAVV RBS compared with the EBOV RBS resulting from the loss of a disulfide bond. A key difference between the RAVV and EBOV RBSs is that Cys121 and Cys147 in the EBOV RBS - which form a disulfide bond - are replaced by Leu105 and His131 in the RAVV RBS, respectively. These substitutions shift the RAVV RBS closer to NPC1-C loop 3, enabling new interactions and establishing loop 3 contacts. (**b**) Structural consequences of the Gly149 (EBOV GPcl) to Gln133 (MBV GPcl)

substitution. The double arrows indicate newly formed interactions between human NPC1 and MBV GPcl due to movement of the MBV RBS loop containing His131-Gln133. (**c**)–(**e**) Comparison of other differences in NPC1-C/RBS interaction networks between EBOV and RAVV GPcls. EBOV GPcl is shown in cyan with its associated NPC1-C in blue, while RAVV GPcl is shown in magenta with its associated NPC1-C in orange. Black double arrows indicate favorable interactions, and red double arrows indicate unfavorable interactions.

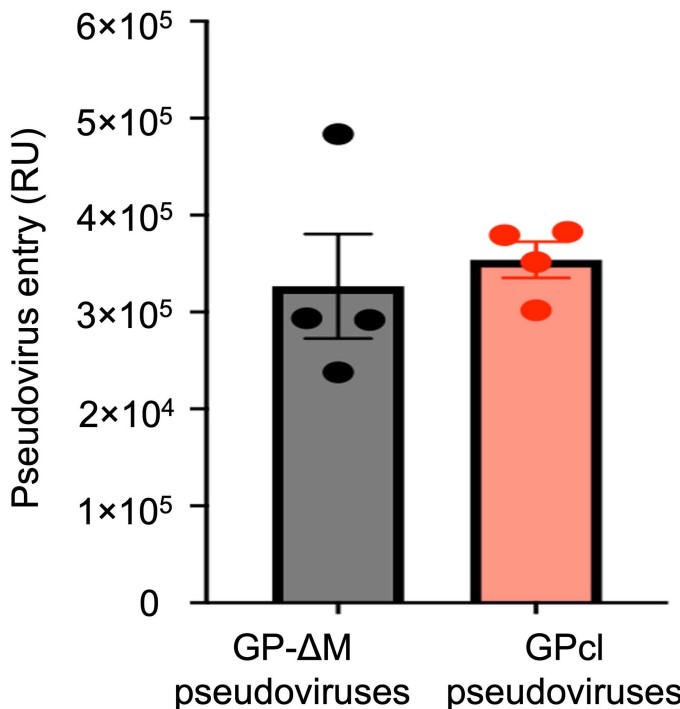

**Extended Data Fig. 9 | Entry of RAVV GP-ΔM and GPcl pseudoviruses into Huh7 cells.** RAVV GP-ΔM pseudoviruses were generated by deleting the mucin-like domain (ΔM) from the RAVV GP construct. RAVV GPcl pseudoviruses were produced by treating RAVV GP-ΔM pseudoviruses with trypsin to remove the glycan cap. Both types of pseudoviruses were then used to infect Huh7 cells. Data are presented as mean ± SEM (n = 4 biologically independent samples).

**Extended Data Table 1 | Cryo-EM data collection, refinement, and validation statistics**

| | RAVV GPcl (PDB 9NPS) (EMD-49631) | RAVV GPcl /NPC1-C (PDB 9NPR) (EMD-49630) | RAVV GP-ΔM /Nanosota-MB1 (PDB 9NPT) (EMD-49632) |
|---|---|---|---|
| **Data collection and processing** | | | |
| Magnification | 130,000 | 130,000 | 130,000 |
| Voltage (kV) | 300 | 300 | 300 |
| Electron exposure (e–/Å²) | 50.00 | 50.00 | 50.00 |
| Defocus range (μm) | -1.0 ~ -2.0 | -1.0 ~ -2.0 | -1.0 ~ -2.0 |
| Pixel size (Å) | 0.664 | 0.664 | 0.664 |
| Symmetry imposed | C1 | C3 | C3 |
| Initial particle images (no.) | 5,365,364 | 3,584,418 | 3,317,427 |
| Final particle images (no.) | 1,083,235 | 116,624 | 176,829 |
| Map resolution (Å) | 3.17 | 3.53 | 2.98 |
| FSC threshold | 0.143 | 0.143 | 0.143 |
| Map resolution range (Å) | 2.6–4.6 | 2.8-6.8 | 2.5-4.5 |
| **Refinement** | | | |
| Initial model used (PDB code) | 6bp2 | 5JQ7 | 6bp2 |
| Model resolution (Å) | 3.4 | 3.8 | 3.2 |
| FSC threshold | 0.5 | 0.5 | 0.5 |
| Model resolution range (Å) | 37.8-1.8 | 43.5-1.3 | 27.2-1.5 |
| Map sharpening $B$ factor (Å²) | -145.5 | -112.2 | -120.5 |
| Model composition | | | |
| Non-hydrogen atoms | 12491 | 11006 | 9375 |
| Protein residues | 1511 | 1311 | 1152 |
| Ligands | 48 | 39 | 24 |
| $B$ factors (Å²) | | | |
| Protein | 118.97 | 76.69 | 61.98 |
| Nucleotide | | | |
| Ligand | 156.49 | 61.75 | 52.59 |
| R.m.s. deviations | | | |
| Bond lengths (Å) | 0.005 | 0.004 | 0.005 |
| Bond angles (°) | 1.018 | 0.806 | 1.021 |
| Validation | | | |
| MolProbity score | 1.47 | 1.84 | 1.57 |
| Clashscore | 3.68 | 8.10 | 4.13 |
| Poor rotamers (%) | 0.15 | 0.95 | 0.62 |
| Ramachandran plot | | | |
| Favored (%) | 95.48 | 94.06 | 94.56 |
| Allowed (%) | 4.39 | 5.78 | 4.46 |
| Disallowed (%) | 0.14 | 0.16 | 0.98 |

See Extended Data Figs. 1–3 for cryo-EM data-processing flowcharts.

# Reporting Summary

## Statistics

For all statistical analyses, confirm that the following items are present in the figure legend, table legend, main text, or Methods section.

| n/a | Confirmed | |
|---|---|---|
| ☐ | ☒ | The exact sample size (*n*) for each experimental group/condition, given as a discrete number and unit of measurement |
| ☐ | ☒ | A statement on whether measurements were taken from distinct samples or whether the same sample was measured repeatedly |
| ☐ | ☒ | The statistical test(s) used AND whether they are one- or two-sided *Only common tests should be described solely by name; describe more complex techniques in the Methods section.* |
| ☒ | ☐ | A description of all covariates tested |
| ☒ | ☐ | A description of any assumptions or corrections, such as tests of normality and adjustment for multiple comparisons |
| ☐ | ☒ | A full description of the statistical parameters including central tendency (e.g. means) or other basic estimates (e.g. regression coefficient) AND variation (e.g. standard deviation) or associated estimates of uncertainty (e.g. confidence intervals) |
| ☐ | ☒ | For null hypothesis testing, the test statistic (e.g. *F*, *t*, *r*) with confidence intervals, effect sizes, degrees of freedom and *P* value noted *Give P values as exact values whenever suitable.* |
| ☒ | ☐ | For Bayesian analysis, information on the choice of priors and Markov chain Monte Carlo settings |
| ☒ | ☐ | For hierarchical and complex designs, identification of the appropriate level for tests and full reporting of outcomes |
| ☒ | ☐ | Estimates of effect sizes (e.g. Cohen's *d*, Pearson's *r*), indicating how they were calculated |

*Our web collection on statistics for biologists contains articles on many of the points above.*

## Software and code

Policy information about availability of computer code

| Data collection | EPU version 2.5. |
|---|---|
| Data analysis | Biacore Evaluation Software v1.1.3; cryoSPARC v4.5.1; MotionCor2; CTFFIND-4.1.13; Coot-0.8.9; Phenix-1.16; UCSF Chimera X v0.93; PyMOL, Version 3.0; LigPlot v.2.2.8; GraphPad Prism v10.4.1; Swiss-Model. |

For manuscripts utilizing custom algorithms or software that are central to the research but not yet described in published literature, software must be made available to editors and reviewers. We strongly encourage code deposition in a community repository (e.g. GitHub). See the Nature Portfolio guidelines for submitting code & software for further information.

## Data

Policy information about availability of data

All manuscripts must include a data availability statement. This statement should provide the following information, where applicable:
- Accession codes, unique identifiers, or web links for publicly available datasets
- A description of any restrictions on data availability
- For clinical datasets or third party data, please ensure that the statement adheres to our policy

The atomic models and corresponding cryo-EM density maps have been deposited into the PDB and the Electron Microscopy Data Bank, respectively, with accession numbers 9NPS and EMD-49631 (RAVV GPcl), 9NPR and EMD-49630 (RAVV GPcl complexed with NPC1-C), 9NPT and EMD-49632 (RAVV GP-ΔM complexed with Nanosota-MB1).

## Research involving human participants, their data, or biological material

Policy information about studies with human participants or human data. See also policy information about sex, gender (identity/presentation), and sexual orientation and race, ethnicity and racism.

| | |
|---|---|
| Reporting on sex and gender | N/A |
| Reporting on race, ethnicity, or other socially relevant groupings | N/A |
| Population characteristics | N/A |
| Recruitment | N/A |
| Ethics oversight | N/A |

Note that full information on the approval of the study protocol must also be provided in the manuscript.

# Field-specific reporting

Please select the one below that is the best fit for your research. If you are not sure, read the appropriate sections before making your selection.

☒ Life sciences       ☐ Behavioural & social sciences       ☐ Ecological, evolutionary & environmental sciences

For a reference copy of the document with all sections, see nature.com/documents/nr-reporting-summary-flat.pdf

# Life sciences study design

All studies must disclose on these points even when the disclosure is negative.

| | |
|---|---|
| Sample size | No formal statistical methods were used to predetermine sample size. Sample sizes were informed by prior experience with SPR binding experiments and pseudovirus entry assays and were chosen to provide independent replication sufficient to assess the reproducibility of the observed effects. |
| Data exclusions | No data points were excluded from the analyses. |
| Replication | The experiments that were repeated, along with the number of replicates, are detailed in the corresponding figure legends. |
| Randomization | Randomization was not performed because this was an observational study. |
| Blinding | Investigators were not blinded to group allocation, as this was an observational study. |

# Reporting for specific materials, systems and methods

We require information from authors about some types of materials, experimental systems and methods used in many studies. Here, indicate whether each material, system or method listed is relevant to your study. If you are not sure if a list item applies to your research, read the appropriate section before selecting a response.

### Materials & experimental systems

| n/a | Involved in the study |
|---|---|
| ☐ | ☒ Antibodies |
| ☐ | ☒ Eukaryotic cell lines |
| ☒ | ☐ Palaeontology and archaeology |
| ☒ | ☐ Animals and other organisms |
| ☒ | ☐ Clinical data |
| ☒ | ☐ Dual use research of concern |
| ☒ | ☐ Plants |

### Methods

| n/a | Involved in the study |
|---|---|
| ☒ | ☐ ChIP-seq |
| ☒ | ☐ Flow cytometry |
| ☒ | ☐ MRI-based neuroimaging |

## Antibodies

| | |
|---|---|
| Antibodies used | Rhodopsin antibody (clone 1D4, sc-57432, Santa Cruz Biotechnology); EBOV GP- and MARV GP–immunized alpaca serum; HRP-conjugated goat anti-alpaca antibody (Jackson ImmunoResearch); HRP-conjugated anti-HA antibody (clone 3F10, 12013819001, |

| Validation | Sigma-Aldrich). |
|---|---|
| | Antibodies were purchased from Santa Cruz Biotechnology and were validated by the manufacturer as documented on their website. Additional information for the rhodopsin antibody (clone 1D4, catalog no. sc-57432) is available in the manufacturer's online datasheet. |

## Eukaryotic cell lines

Policy information about cell lines and Sex and Gender in Research

| Cell line source(s) | HEK293T (ATCC, American Type Culture Collection), Huh7 (ATCC), THP-1 (ATCC), HUVEC (ATCC), and Expi293F (Thermo Fisher Scientific) cells. |
|---|---|
| Authentication | HEK293T, Huh7, and THP-1 cells were authenticated by the vendors using STR profiling. Authentication documentation for Expi293F and HUVEC cells was not available on the vendors' websites. |
| Mycoplasma contamination | HEK293T cells were tested for mycoplasma contamination in our laboratory and by the vendor and were negative in both cases. Huh7, HUVEC, THP-1, and Expi293F cells were tested and confirmed mycoplasma-negative by the vendors. |
| Commonly misidentified lines (See ICLAC register) | None reported. |

## Plants

| Seed stocks | N/A |
|---|---|
| Novel plant genotypes | N/A |
| Authentication | N/A |

