## [Peer Review file · Nature]

Structures of Marburgvirus Glycoprotein and Its Complex with NPC1 Receptor

Corresponding Author: Professor Fang Li

Version 0:

Reviewer comments:

Referee #1

(Remarks to the Author)

Ye et al. determined the cryoEM structures of Marburgvirus (MBV) glycoproteins in three states: unbound, bound to NPC1 (a receptor in the endosome), and complexed with a nanobody. They compared the structures to the well defined EBOV glycoprotein. Results explain why MBV is more efficient in infection compared to EBOV. The structural difference involved the glycan cap of MBV shields the RBS from NPC1 but only partially from nanobody while that of EBOV is shield from both. There are structural changes induced by NPC1 binding which might be involved in facilitating fusion. They identified a nanobody that binds to a similar site as NPC1 but with high affinity, indicating its potential use as a therapeutic agent. The manuscript is well written and is easy to read.

This reviewer thinks the major findings of this manuscript is the structures of MBV glycoprotein in unbound and bound form with NPC1. Regarding nanobodies, there are other human antibodies that bind to similar sites as discussed by the authors. However, nanobodies are more stable and thus could be a better therapeutic agent than human antibodies. Regarding the inference to fusion, the claim about "significant" structural changes is not that "significant" compared to class I or class II fusion viruses.

Major comments:

- (1) Since NPC1 binds to MBV in the endosome, the pH environment should be acidic. Authors wrote in the method section that they added CHAPSO before freezing, but did not indicate what pH. If done in neutral pH, should authors conduct the same experiment at low pH? - to determine the actual structure that might correlate more to the biological event. Maybe they would then see more drastic fusion transitional structural changes?
- (2) The authors should discuss the fusion difference compared to class I, II and III fusion proteins.

Minor comments:

- (1) Page 4, line 75: change to "much less characterized".
- (2) Page 9, line 193 to 199, in the paragraph, authors refer to N-linked glycan at residue 84 (N84 glycan) but in fig. S4 and S8, they are labeled as N94.
- (3) Page 9, line 201, Kd figure point to Fig 2C but it is actually in Fig.2C.
- (4) This reviewer is not familiar with the field, can author define what is apo-structure?- unbound?

Referee #2

(Remarks to the Author)

A. Summary of the key results

Ye and Bu et al. present a study on the structures of Ravn virus glycoprotein and its interactions with the NPC1 receptor and a neutralizing nanobody. The paper aims to understand the molecular mechanism behind the enhanced entry of Marburgviruses. Since Marburg virus glycoprotein had poor yields, the authors primarily worked with the closely related Ravn virus. The study provides critical structural insights, and conceptual mechanistic novelty that inform future antiviral development against this class of virus.

B. Originality and significance

This is original work and all structures and the nanobody are novel. The findings are of high significance to develop effective

antiviral strategies against Marburgviruses, and may contribute to a better understanding of the cost-benefit relationship between RBS accessibility and shielding.

C. Data & methodology: validity of approach, quality of data, quality of presentation

This paper is overall very well done. Solid data, relevant experiments, clear presentation, logical flow of information. While the authors claim to have identified a neutralising nanobody, their data only supports pseudotyped virus neutralisation using a nanobody-Fc fusion. The entire text (excluding methods and figure captions) does not mention the use of a Fc-fusion. Instead of a neutralizing nanobody, the authors identified a neutralising heavy-chain only antibody. I understand that this virus presents challenges, but these important details must be clearly stated, perhaps already in the title. Nobody would claim identification of a neutralizing Fab-fragment, but only show full-sized antibody data.

D. Appropriate use of statistics and treatment of uncertainties

Yes.

E. Conclusions: robustness, validity, reliability

The study appears to be solid, except for points mentioned in section C and F.

F. Suggested improvements: experiments, data for possible revision

The authors present a nanobody that can bind the RBS of RAVV. It can even bind GP- Δ M, albeit at much lower affinity. That is a fantastic finding! However, the high affinity to a physiologically irrelevant version of a viral protein is meaningless. But that's OK as long as we can learn from it.

The authors cannot find the glycan cap loop in their structures. Because of its absence, the authors conclude that it may regulate access to the RBS. In that case, the cap should be there in the absence of NPC1 or nanobody. The authors can make their (important) statement, if they provide the data.

Alternatively, the authors should provide any (structural?) data explaining the significant difference of nanobody affinity to GP- Δ M and GPcl.

This would be meaningful in inspiring future, more potent targeted therapies that more efficiently bypass the glycan cap loop and neutralise MBV at much lower concentrations.

G. References: appropriate credit to previous work?

Yes. However, the authors like to cite themselves. While they did great work previously, I recommend to only include citations that are actually meaningful to this paper.

H. Clarity and context: lucidity of abstract/summary, appropriateness of abstract, introduction and conclusions

The abstract ends with "highlighting nanobodies as a promising therapeutic strategy against MBV infections", yet:

- 1) the authors do not provide any neutralization data with authentic virus,
- 2) there is no in vivo challenge data,
- 3) there is no data on developability of the nanobody, or the Fc-fusion,
- 4) and as Fc-fusion the nanobody only has an IC50 of $>0,5$ μ g/ml.

In summary, this protein is far away from a therapy. I suggest that the authors instead focus more on their impressive biological and conceptual findings, especially in the abstract.

The authors write: "We have recently developed nine nanobodies targeting SARS-CoV-2 spike protein and two more targeting EBOV GP". I am sincerely happy about their previous success, but as it is, this sentence does not add meaning to this paper. Please indicate how the previous studies have impacted or guided biological findings, or otherwise, delete this section.

Line 75 "Compared to EBOV GP, MBV GPs are much less uncharacterized." It is possible that the authors mean "characterized".

Line 110 Readers might appreciate details on the strains of MARV that the authors encountered difficulties producing recombinantly. It would also increase consistency with figures 5E and S5.

It would be helpful to mention in the main text, and refer to Fig S5, how similar the RBS of RAVV and MARV are.

Referee #4

(Remarks to the Author)

Ye and colleagues determined the structure of the Ravn virus (RAVV) virus glycoprotein (GP) without mucin-like domain (MLD) and glycan cap (RAVV-GPcl), RAVV-GPcl in complex with NPC1 and RAVV-delta MLD in complex with a nanobody. In brief, the study indicates limited conformational flexibility of the RAVV-GP glycan cap, substantial differences in the spatial orientation of NPC1 needed for EBOV- and MARV-GP binding, binding of three NPC1 loops to the RAVV-GP receptor binding site (RBS) as compared to two interacting with EBOV-GP RBS, reveals structural changes in GP associated with receptor binding and confirms and extends the RBS as a target for antibodies and related biologicals. These findings are of significant interest. However, several points remain to be addressed.

Major

“In addition to their higher fatality rate, MBVs also exhibit greater cell infectivity than EBOV 9,10.” References 9 and 10 do not adequately support this statement since neither pseudotypes nor authentic filoviruses were normalized for capsid/VP40 and GP content. Similarly, the statement in the present manuscript “demonstrating that MBV GPs guide viral entry more effectively than EBOV GP” is not adequately supported by data. Thus, more cell lines and, importantly, more relevant cell lines need to be analyzed, including cell lines mimicking macrophages (for instance THP-1 derived macrophages), the major viral target cell type. Further, the following statement in the figure legends requires revision: “Pseudovirus entry signals were normalized based on GP expression levels in each pseudovirus, as determined by Western blot targeting the C9 tag of the GPs. Pseudovirus levels were also assessed by Western blot targeting the retroviral capsid protein P24. Data are presented as mean \pm SEM (n = 24)”. Does this mean that the y-axis of figure 1D does not show unprocessed light units but light units relative to p24 and GP levels? If so, the unprocessed data must also be shown and it must be indicated how data were normalized. Further, N = 24 is unusual. It is important to state how many biological replicates were averaged and how many technical replicates were analyzed per biological replicate. Please also indicate how many separate pseudotype stocks were analyzed. Finally, data should be confirmed with untagged GPs since it cannot be excluded that the C9 tag interferes with cell entry driven by EBOV-GP but not Marburg virus glycoproteins.

The study indicates that NPC1 residues D436 and I437 interact with the GPs of Marburg viruses but not EBOV. It is important to examine the significance of these residues for EBOV- relative to MARV- and RAVV-GP-driven entry. Thus, one would expect that mutating these NPC1 residues reduces MARV- and RAVV- but not EBOV-GP-driven entry.

The authors suggest that N-glycans attached to RAVV-GP N94 and NPC1 N557 interact and that this interaction promotes cell entry driven by Marburg virus glycoproteins. This conclusion must be supported by additional data: It should be ensured that the reduced entry of RAVV-GP bearing particles upon mutation of N94 is not due to reduced attachment to target cells. Further, it should be tested whether mutation of N557 selectively interferes with entry driven by Marburg virus glycoproteins but not EBOV-GP.

The binding affinity of RAVV-GPcl was determined and compared with that published for EBOV-GPcl. However, in order to support statements like “.., which is significantly stronger than the binding affinity of EBOV GPcl (Kd = 20.4 μ M)” the binding affinity of EBOV-GPcl must be measured by the authors.

Does exposure of trypsin-treated particles bearing RAVV-GP without MLD to recombinant NPC1 trigger membrane fusion?

The implications of these findings for efforts to target NPC1 for anti-filovirus therapy should be discussed.

Minor

“The average case fatality rate of MBVs exceeds 70% and can reach ~90% in some outbreaks - significantly higher than the ~50% fatality rate of two prevalent ebolavirus species, Ebola virus (EBOV) and Sudan virus (SUDV) 5-8.” References must be included that list the number of all previous cases of infection with EBOV, SUDV and Marburg viruses and the ensuing deaths.

The text discusses N84 while the residue in the figure is labelled as N94.

Please provide details in the methods section how recombinant NPC1 was prepared.

Referee #5

(Remarks to the Author)

Ye and co-authors present a new structure of Marburg virus (MARV) GP bound to domain C of the filovirus receptor NPC1. This structure points to some interesting features of the MARV GP apo structure that had not yet been uncovered, including the retention of a glycan cap sequence in the receptor-binding cleft. The GPcl: NPC1 structure reveals an apparent rotation of the interface relative to that of EBOV, and the involvement of distinct NPC1 residues in the interaction. The authors also report the structure of a new VHH (nanobody) bound into the receptor-binding pocket, which is rather reminiscent of how a class of human mAbs (e.g., MR72 and MR191) recognize cleaved MARV GP and block receptor binding.

Specific comments:

1. Little functional data is presented to buttress the new GPcl-NPC1 structure and the authors' inferences from it regarding the mode of MARV GP-NPC1 interaction. At minimum, they should perform mutagenesis on the new contact residues in NPC1 they have identified and assess the binding of these mutants to EBOV and MARV GPcl.

2. There is a body of literature from multiple groups describing MARV GP and NPC1 mutagenesis, the mode of GP-NPC1 interaction, NPC1 ortholog- and filovirus GP-dependent similarities and differences in this interaction for ebolaviruses and marburgviruses,—a partial list below: The authors should discuss their structure in light of this work.

Lasso G et al., 2025, PMID: 39818205
Takada A et al., 2020, PMID: 31940478
Takada A et al., 2018, PMID: 30010949

Bornholdt Z et al., 2015, PMID: 26908579
Ng M et al., 2015, PMID: 26698106
Manicassamry B et al., 2007, PMID: 16989883

3. The claim that MARV GP mediates entry more efficiently than EBOV GP is unconvincing, especially given the data presented. For instance, the authors are looking at viral entry using a cell line—any differences they see could just be an artifact of the specific system they are using. If they believe that the higher GPcl-NPC1 binding affinity affords higher entry efficiency, they should support that hypothesis with data. For example, they could mutate MARV GP to reduce its affinity to a similar level as that of EBOV GP and then look at viral infectivity. It is likely that there is not a linear relationship between binding affinity and entry efficiency, especially given the likely avid interaction between virion-bound GPcl and NPC1 in late endosomes and lysosomes.

The idea that the presented differences in entry affect virulence is even more of a stretch, given the many substantial biological differences between ebolaviruses and marburgviruses. In this reviewer's opinion, these experiments are tangential to the main thrust of the paper, which reports new structures. The authors would be better served focusing on the mechanism of the GP:NPC1 interaction and its direct biological implications.

Version 1:

Reviewer comments:

Referee #1

(Remarks to the Author)

This reviewer has no additional comments.

Referee #2

(Remarks to the Author)

In this revised version, Ye and Bu et al. have substantially improved the initial manuscript and addressed the previous concerns. However, I have two additional points that should be clarified or further addressed.

1. In the revision, the authors include tagged GPs to normalize pseudotyped viruses. They assume that by normalizing for expression levels, any observed differences in RUs must reflect differences in viral entry efficiency. However, expression levels alone do not necessarily correlate with production of functional pseudotyped viruses. Other mechanisms could also contribute to the observed phenotypes. For example, variations in the proportion of functional GPs produced in 293T cells, or differences in the efficiency with which these GPs are incorporated into viral particles. The authors should discuss these possibilities.

2. The authors refer to K_d as affinity in Figure 2, while in Figure 3, they use K_a . They likely mean K_D , which is defined by k_d/k_a . The authors should clarify whether the data presented corresponds to association (k_a) or dissociation (k_d) rates, explain the rationale for using different terms or units across figures, and ensure consistent nomenclature throughout.

Referee #4

(Remarks to the Author)

The authors have adequately addressed most of my points. Considering that conclusions based on structural information should be confirmed by functional studies, it should also be examined whether mutating NPC1 residues D436 and I437 not only reduces MARV-GP but not EBOV-GP binding but also diminishes MARV-GP but not EBOV-GP-driven cell entry.

Referee #5

(Remarks to the Author)

Overall, the most interesting and important aspect of this work, in my opinion, is the evidence for a novel mode of NPC1 binding by MARV GPcl relative to EBOV GPcl. Please see below for some specific comments related to this. I feel that this is the real strength of this manuscript. By comparison, the structure with the VHH, while interesting, is less noteworthy to me. Thus, I am also less concerned about the in vivo data with the VHH (or lack thereof).

To me, the claim that the enhanced binding affinity of MARV GPcl to NPC1-C accounts for its increased entry activity still seems like a reach given the complexity of the biology at both the single-cell and organismal level. I feel the authors' emphasis of this points detracts from the manuscript. In my opinion, the conclusions from these experiments should be greatly de-emphasized.

Some specific comments:

Broadly speaking, the authors could do much more in the text and figs to compare and contrast the binding modes of EBOV

and MARV GP to NPC1, given that this is the main take-home message of the paper.

- Lines 165-169: To facilitate readership, please specify the residue numbers comprising each NPC1 loop when first describing the loops mediating the interaction with RAVV GPcl.
- Lines 180-183: Mutagenesis experiments show that residues in loop 3 are important but I would not say they are critical since the mutant NPC1 still binds (K_a increases by less than a log). I suggest lowering the tone of the statement.
- Line 205-207: K155 and D502 are too far (4.7Å) to form a salt-bridge in EBOV GP – NPC1 complex (pdb: 5f1b). Also, F502 in NPC1 doesn't seem to clash with K155 in EBOV GP if I perform in-silico mutational analysis on the EBOV GP – NPC1 complex. I would suggest revisiting this section of the ms and include intermolecular distance measures in the corresponding figures. Are L139 and F205 too far from a favorable hydrophobic contact?
- Line 210: What is the area of the binding pocket in RAVV GPcl compared to EBOV GPcl? This would help further supporting the idea that Q133 contributes to the opening of the pocket. However, it is possible that other residues differences contribute too (e.g. the lack of disulfide bond between 121-147 cysteines -EBOV numbering). I strongly suggest that the authors compare the geometry of the GP1 pocket in EBOV and RAVV. Along these lines, the positioning of the beta sheets on one side of the RAVV GP1 (residues 92-98 & 121-128) pocket is remarkably different from that in EBOV GP1, narrowing the opening on the pocket. I didn't see any comment on this in the manuscript.
- Could the authors speculate in their discussion what factors might drive the different binding mode of RAVV GPcl? Is the shape of the RAVV GP binding pocket driving the binding mode? This is to me a very interesting finding that would benefit from some additional explanation.
- Lines 292-294. The sentence implies that loop 3 is particularly relevant for the ~11-fold higher binding affinity to human NPC1 than EBOV GPcl. However, the experimental data suggest that the interactions driven by H418 and I419 are more relevant. These mutagenesis experiments assess the individual contribution of each residue separately, but it does not address potential epistatic effects. I would suggest caution when describing the importance of loop 3. While important, the experimental data does not support that these are critical residues for binding.
- Include distances in the figures that describe molecular interactions: Figs 5b, S7, S8, S12 & S13.

Responses to Editor and Reviewers' comments:

Editor

(1) Review: The reviewers find the GP-NPC1 structure interesting, but Reviewer 1 raises technical concerns regarding whether the structure is representative of the conformation at acidic pH.

Response: The structure was indeed determined at acidic pH. At neutral pH, GP-NPC1 binding is much weaker; therefore, both our SPR and cryo-EM studies were carried out under acidic conditions. We have now provided the detailed cryo-EM buffer conditions in the revised manuscript to clarify this point.

(2) Review: The reviewers consider the nanobody interesting but feel it does not represent a sufficient advance, as its therapeutic activity has not been tested in vivo (and similar antibodies already exist).

Response: While the manuscript was under review, we obtained in vivo data showing that the nanobody effectively protects mice from MBV infection. However, following the editor's and reviewers' suggestions, we have focused the current study on MBV entry. This focus allows us to provide a more coherent and in-depth structural and mechanistic analysis without overextending into therapeutic claims. Accordingly, we removed statements about nanobody therapeutic efficacy from the abstract and discussion and toned down implications regarding therapy. Instead, we added the following sentence to the discussion to clarify the role of the nanobody in this study:

“While employed here primarily as a research tool to probe MBV entry, Nanosota-MB1 also shows potential as a neutralizing therapeutic against MBVs.”

(3) Review: The proposed mechanistic differences between EBOV and MARV entry have not been further explored or functionally validated (e.g., by mutagenesis or analysis in more relevant cell lines).

Response: In the revised manuscript, we have added several lines of functional validation:

- (i) Expanded pseudovirus entry assays. We tested viral entry across three relevant human cell types—Huh7 (human hepatoma cell line), HUVEC (primary human vascular endothelial cells), and THP-1-derived macrophages - all major cellular targets of filoviruses. EBOV pseudoviruses were included for direct comparison. RAVV and Musoke pseudoviruses entered these cells far more efficiently than EBOV pseudoviruses (>300-, 25-, and 12-fold in three cell types, respectively), demonstrating that MBV GPs are substantially more effective entry mediators (Fig. 1D; Fig. S4).
- (ii) Comparative receptor binding analysis. Using SPR, we directly compared the binding affinities of RAVV and EBOV GPc1s, performing side-by-side measurements in triplicate for statistical robustness. RAVV GPc1 bound NPC1-C ~11-fold more strongly than EBOV GPc1 (Kd values of 394 nM and 4.34 μ M, respectively) (Fig. 3D, 3E).
- (iii) Mutational analysis of NPC1-C loop 3. To test the structural role of loop 3, we substituted Asp436 and Ile437 with alanine and measured GPc1 binding by SPR. These substitutions markedly reduced RAVV GPc1 affinity but had little effect on EBOV GPc1, confirming that loop 3 specifically stabilizes RAVV recognition (Figs. S7C, S9–S11).
- (iv) Mutational analysis of additional NPC1-C residues. We also introduced mutations at other NPC1-C residues that differentially interact with RAVV and EBOV GPc1s, with results consistently supporting our structural analysis.

Conclusion: Together, these differences - particularly the new loop 3 contacts - explain why RAVV GPcl binds NPC1-C with substantially higher affinity than EBOV GPcl.

(4) In addition to the above developments of the study, we have developed a novel method to compare the entry efficiencies of two viral glycoproteins, which was previously not possible. Please see the updated results below:

“A long-standing challenge has been the lack of a reliable method to directly compare viral entry mediated by different filovirus GPs when normalized to equivalent expression levels. This difficulty arises because no neutralizing sera or antibodies recognize EBOV and MBV GPs equivalently, and the use of a common artificial tag risks interfering with entry. To address this, we developed a novel double-normalization strategy (Fig. S4). Specifically, we generated two versions of each GP: wild type and C-terminally tagged with C9. We then measured entry efficiency for six pseudoviruses: EBOV, RAVV, and Musoke, each with or without the C9 tag. In the first step, wild-type EBOV, RAVV, and Musoke pseudoviruses were normalized to their respective C9-tagged counterparts using neutralizing sera specific to each GP. In the second step, C9-tagged RAVV and Musoke pseudoviruses were further normalized to C9-tagged EBOV pseudoviruses using anti-C9 antibodies. This two-step procedure ensured that all six pseudoviruses were standardized to equivalent GP expression, enabling direct comparison of viral entry mediated by the three wild-type GPs.”

Reviewer 1

(1) Review: Since NPC1 binds to MBV in the endosome, the pH environment should be acidic. Authors wrote in the method section that they added CHAPSO before freezing, but did not indicate what pH. If done in neutral pH, should authors conduct the same experiment at low pH? - to determine the actual structure that might correlate more to the biological event. Maybe they would then see more drastic fusion transitional structural changes?

Response: Please see our response to Editor comment #1, where we clarify that all structural and binding experiments were performed at acidic pH conditions mimicking the endosomal environment.

(2) Review: The authors should discuss the fusion difference compared to class I, II and III fusion proteins.

Response: We have added the following comparison:

“Filovirus GPs belong to the class I family of membrane fusion proteins, whose prefusion trimers collapse into a post-fusion six-helix bundle. This mechanism differs from that of class II proteins (e.g., dengue virus E), which transition from prefusion dimers to elongated β -sheet trimers, and class III proteins (e.g., herpesvirus gB), which transition from prefusion trimers to post-fusion trimers with mixed class I/II features³².”

(3) Review: Page 4, line 75: change to “much less characterized”.

Response: This has been corrected to ‘much less characterized’ as suggested.

(4) Review: Page 9, line 193 to 199, in the paragraph, authors refer to N-linked glycan at residue 84 (N84 glycan) but in fig. S4 and S8, they are labeled as N94.

Response: It should be N94. This has been corrected throughout the text and figures as suggested.

(5) Review: Page 9, line 201, Kd figure point to Fig 2C but it is actually in Fig.2C.

Response: In the original submission, Page 9, line 201, the Kd figure was incorrectly referenced as Fig. 2D. This has now been corrected to Fig. 2C, which is the accurate reference.

(6) Review: This reviewer is not familiar with the field, can author define what is apo-structure?- unbound?

Response: Yes. An apo structure refers to the protein in its unbound state (i.e., without ligand or binding partner).

Reviewer 2

(1) Review: While the authors claim to have identified a neutralising nanobody, their data only supports pseudotyped virus neutralisation using a nanobody-Fc fusion. The entire text (excluding methods and figure captions) does not mention the use of a Fc-fusion. Instead of a neutralizing nanobody, the authors identified a neutralising heavy-chain only antibody. I understand that this virus presents challenges, but these important details must be clearly stated, perhaps already in the title. Nobody would claim identification of a neutralizing Fab-fragment, but only show full-sized antibody data.

Response: In the revised manuscript, we now explicitly state in the Results and Discussion sections that Fc-tagged Nanosota-MB1 was used in the pseudovirus neutralization assays. In addition, we clarified in the Methods the rationale for including the Fc tag:

“The Fc-tag was included to enhance nanobody multivalency for GP interactions and to increase in vivo half-life, while preserving the single-domain structure for antigen binding. The resulting construct remains approximately half the size of IgGs, making it compatible with intranasal administration.”

These changes ensure that the distinction between the nanobody and its Fc-fusion format is clear throughout the manuscript.

(2) Review: The authors present a nanobody that can bind the RBS of RAVV. It can even bind GP-ΔM, albeit at much lower affinity. That is a fantastic finding! However, the high affinity to a physiologically irrelevant version of a viral protein is meaningless. But that’s OK as long as we can learn from it.

The authors cannot find the glycan cap loop in their structures. Because of its absence, the authors conclude that it may regulate access to the RBS. In that case, the cap should be there in the absence of NPC1 or nanobody.

The authors can make their (important) statement, if they provide the data.

Alternatively, the authors should provide any (structural?) data explaining the significant difference of nanobody affinity to GP-ΔM and GPc1.

This would be meaningful in inspiring future, more potent targeted therapies that more efficiently bypass the glycan cap loop and neutralise MBV at much lower concentrations.

Response: In the apo structure of RAVV GPc1, we detected the glycan cap loop despite complete proteolytic removal of the cap, indicating that the loop is not entirely flexible as previously suggested. Our SPR data further support this, showing that the glycan cap only partially restricts nanobody access to the RBS. Importantly, even with the cap present, the nanobody binds GP-ΔM with a Kd of 4.77×10^{-8} M, which still reflects strong affinity. Thus, the GP-ΔM binding result is not simply an artifact of a “physiologically irrelevant” construct, but instead

highlights how partial flexibility of the glycan cap allows nanobody access to the RBS even in the prefusion state.

These findings provide mechanistic insight into MBV entry and neutralization. As discussed in the manuscript:

- (i) “This raises an evolutionary question: what advantage does RAVV gain from a glycan cap with partial flexibility, allowing only limited immune evasion? One possibility is that increased flexibility facilitates glycan cap removal in endosomes, thereby enhancing NPC1 binding and viral entry. Indeed, prior work - including ours - has shown that stabilizing the glycan cap impairs proteolysis, reduces RBS exposure, and inhibits EBOV GP-mediated entry^{26,33}. If the same applies to MBV GP, this would suggest that RAVV has evolved a partially flexible glycan cap to balance immune evasion with efficient entry.”
- (ii) “Moreover, due to the partial flexibility of the glycan cap, Nanosota-MB1 can bind MBV GP with high affinity even before endocytosis, further enhancing its efficacy.”

(3) Review: However, the authors like to cite themselves. While they did great work previously, I recommend to only include citations that are actually meaningful to this paper.

Response: In the revised manuscript, we have removed less relevant citations, including our prior work on Sudan filovirus GP and several publications on nanobodies targeting SARS-CoV-2. We have retained only references that are directly relevant to the present study, as recommended.

(4) Review: The abstract ends with “highlighting nanobodies as a promising therapeutic strategy against MBV infections”, yet:

- 1) the authors do not provide any neutralization data with authentic virus,
- 2) there is no in vivo challenge data,
- 3) there is no data on developability of the nanobody, or the Fc-fusion,
- 4) and as Fc-fusion the nanobody only has an IC₅₀ of >0,5 ug/ml.

In summary, this protein is far away from a therapy. I suggest that the authors instead focus more on their impressive biological and conceptual findings, especially in the abstract.

Response: Please see our response to Editor Comment #2. Although we have obtained in vivo data on the nanobody, we have chosen to follow the editor’s and reviewer’s recommendation to focus this study on MBV entry. In this context, we use the nanobody primarily as a research tool to study the structure and function of MBV GP, while only briefly mentioning its potential therapeutic implications. We anticipate that future studies can more fully explore its therapeutic potential, including neutralization with authentic virus and in vivo efficacy.

(5) Review: The authors write: “We have recently developed nine nanobodies targeting SARS-CoV-2 spike protein and two more targeting EBOV GP”. I am sincerely happy about their previous success, but as it is, this sentence does not add meaning to this paper. Please indicate how the previous studies have impacted or guided biological findings, or otherwise, delete this section.

Response: We have removed this section as suggested, since it is not directly relevant to the present study.

(6) Review: Line 75 “Compared to EBOV GP, MBV GPs are much less uncharacterized.” It is possible that the authors mean "characterized".

Response: This was a typographical error and has been corrected.

(7) Review: Line 110 Readers might appreciate details on the strains of MARV that the authors encountered difficulties producing recombinantly. It would also increase consistency with figures 5E and S5.

Response: We have clarified this in the revision:

“The RAVV GP ectodomain exhibited significantly higher expression yield and stability than MARV GP ectodomains from the Musoke and Angola strains, making it the preferred candidate for cryo-EM analysis.”

Reviewer 4

(1) Review: “In addition to their higher fatality rate, MBVs also exhibit greater cell infectivity than EBOV 9,10.” References 9 and 10 do not adequately support this statement since neither pseudotypes nor authentic filoviruses were normalized for capsid/VP40 and GP content.

Response: We have removed this sentence along with the two associated references. In addition, all comparisons of viral infectivity have been eliminated. The revision now focuses exclusively on comparing the cell entry efficiencies mediated by MBV and EBOV GPs.

(2) Review: Similarly, the statement in the present manuscript “demonstrating that MBV GPs guide viral entry more effectively than EBOV GP” is not adequately supported by data. Thus, more cell lines and, importantly, more relevant cell lines need to be analyzed, including cell lines mimicking macrophages (for instance THP-1 derived macrophages), the major viral target cell type.

Response: Please see our response to Editor Comment #2. Briefly, we expanded pseudovirus entry assays to include three relevant human cell types—Huh7, HUVEC, and THP-1—derived macrophages—all major cellular targets of filoviruses. Compared with EBOV pseudoviruses, RAVV and Musoke pseudoviruses entered these cells far more efficiently (>300-, 25-, and 12-fold in three cell types, respectively), demonstrating that MBV GPs mediate entry substantially more effectively (Fig. 1D; Fig. S4).

(3) Review: Further, the following statement in the figure legends requires revision: “Pseudovirus entry signals were normalized based on GP expression levels in each pseudovirus, as determined by Western blot targeting the C9 tag of the GPs. Pseudovirus levels were also assessed by Western blot targeting the retroviral capsid protein P24. Data are presented as mean \pm SEM (n = 24)”. Does this mean that the y-axis of figure 1D does not show unprocessed light units but light units relative to p24 and GP levels? If so, the unprocessed data must also be shown and it must be indicated how data were normalized. Further, N = 24 is unusual. It is important to state how many biological replicates were averaged and how many technical replicates were analyzed per biological replicate. Please also indicate how many separate pseudotype stocks were analyzed.

Response: Please see our response #4 to the Editor. The reviewers’ comments inspired us to develop a novel double-normalization method to directly compare the entry efficiencies of two viral glycoproteins, which was previously not possible. In addition to Fig. 1D–F, which show the fully normalized results for the three wild-

type filovirus GPs, we have included a new Fig. S4 to detail the normalization procedure and address the reviewer's concerns. Specifically:

“Fig. S4. Entry efficiencies of filovirus pseudoviruses into three human target cell types, measured using a double-normalization strategy to control for GP expression levels. Retroviruses pseudotyped with EBOV, RAVV, or Musoke GPs were used to infect Huh7 (hepatoma), HUVEC (primary vascular endothelial), and THP-1–derived macrophages - all major cellular targets of filoviruses. For each GP, both wild-type and C-terminally C9-tagged versions were packaged, yielding six pseudoviruses in total. Entry efficiencies were measured in four biological replicates per pseudovirus and cell type. Normalization was performed in two steps: first, wild-type pseudoviruses were standardized to their C9-tagged counterparts using GP-specific neutralizing sera; second, C9-tagged RAVV and Musoke pseudoviruses were normalized to C9-tagged EBOV pseudoviruses using anti-C9 antibodies. This double-normalization ensured equivalent GP expression levels across all pseudoviruses, allowing direct comparison of entry efficiencies mediated by the three wild-type GPs. All experiments were repeated three times with independently prepared pseudovirus stocks, yielding consistent results.”

(4) Review: Finally, data should be confirmed with untagged GPs since it cannot be excluded that the C9 tag interferes with cell entry driven by EBOV-GP but not Marburg virus glycoproteins.

Response: Please see our response #4 to the Editor. Thank you for this excellent suggestion. To directly address this concern, we included both wild-type (untagged) and C-terminally C9-tagged versions of each GP in our experiments (see new Fig. S4). Entry efficiencies were consistent between the wild-type and tagged constructs, both confirming that MBV GPs mediate entry more efficiently than EBOV GP, although the C9 tag did partially interfere with entry driven by EBOV and MARV GPs. Importantly, your comment also inspired us to develop a novel double-normalization method, which allows direct comparison of entry efficiencies across different filovirus GPs at equivalent expression levels—something that was not previously possible. We believe this approach represents a significant methodological advance with broad applicability in virology.

(5) Review: The study indicates that NPC1 residues D436 and I437 interact with the GPs of Marburg viruses but not EBOV. It is important to examine the significance of these residues for EBOV- relative to MARV- and RAVV-GP-driven entry. Thus, one would expect that mutating these NPC1 residues reduces MARV- and RAVV- but not EBOV-GP-driven entry.

Response: We have conducted this experiment and incorporated the results into the updated manuscript:

“A key difference is that Cys121 and Cys147 in the EBOV RBS - linked by a disulfide bond - are replaced by Leu105 and His131 in the RAVV RBS (Fig. S8). These substitutions shift the RAVV RBS closer to loop 3 of NPC1-C, enabling new interactions with Ile437 and Lys498 of NPC1-C and thereby establishing loop 3 contacts (Fig. S8). These contacts specifically involve Asp436 and Ile437 of NPC1-C (Fig. S7C). Alanine substitutions at these two positions markedly reduced RAVV GPcl binding affinity (Fig. S9–S11), confirming the critical role of loop 3 in RAVV recognition. By contrast, the same substitutions had little effect on EBOV GPcl binding, consistent with the absence of loop 3 contacts in EBOV (Fig. S9–S11).”

(6) Review: The authors suggest that N-glycans attached to RAVV-GP N94 and NPC1 N557 interact and that this interaction promotes cell entry driven by Marburg virus glycoproteins. This conclusion must be supported by additional data: It should be ensured that the reduced entry of RAVV-GP bearing particles upon mutation of

N94 is not due to reduced attachment to target cells. Further, it should be tested whether mutation of N557 selectively interferes with entry driven by Marburg virus glycoproteins but not EBOV-GP.

Response: We performed the requested experiments; however, mutating either glycan (RAVV GP N94 or NPC1 N557) resulted in aggregation of the corresponding recombinant proteins. This indicates that these glycans are critical for proper folding. Consequently, neither SPR binding assays nor pseudovirus entry assays could be reliably performed. We have therefore revised our conclusion as follows:

“In addition to these amino acid differences, RAVV GP contains an N-linked glycan at residue 94 (N94 glycan) (Fig. S5C), which is absent in EBOV GP. The N94 glycan of RAVV GP interacts with the N557 glycan of NPC1-C. However, mutating either glycan caused aggregation of recombinant RAVV GPcl or NPC1-C, preventing further functional analysis. As such, it remains unclear whether the N94 glycan enhances or reduces NPC1 binding affinity, or whether it modulates GP interactions with cell-surface attachment factors (e.g., lectins) during viral endocytosis.”

(7) Review: The binding affinity of RAVV-GPcl was determined and compared with that published for EBOV-GPcl. However, in order to support statements like “.., which is significantly stronger than the binding affinity of EBOV GPcl ($K_d = 20.4 \mu\text{M}$)” the binding affinity of EBOV-GPcl must be measured by the authors.

Response: Using SPR, we directly measured and compared the NPC1-C-binding affinities of RAVV and EBOV GPcls in side-by-side experiments. Measurements were performed in triplicate to ensure statistical robustness. RAVV GPcl bound NPC1-C ~11-fold more strongly than EBOV GPcl, with K_d values of 394 nM and 4.34 μM , respectively (Fig. 3D, 3E). Full SPR sensorgrams from all replicates are provided in Fig. S10 and S11.

(8) Review: Does exposure of trypsin-treated particles bearing RAVV-GP without MLD to recombinant NPC1 trigger membrane fusion?

Response: We performed pseudovirus entry assays using three versions of RAVV GP: full-length GP (Fig. 1D–F), GP lacking the mucin-like domain (GP- ΔM , Fig. S14A), and GPcl (Fig. S14B). All three constructs mediated efficient entry, indicating that each can be triggered by NPC1 to promote membrane fusion.

(9) Review: The implications of these findings for efforts to target NPC1 for anti-filovirus therapy should be discussed.

Response: We have added the following discussion to address this comment:

“The multiple roles of NPC1 in RAVV entry, including its high-affinity binding to GPcl and its ability to trigger conformational change, underscore the GPcl–NPC1 interface as a key target for antiviral inhibitors acting on either GPcl or NPC1.”

(10) Review: “The average case fatality rate of MBVs exceeds 70% and can reach ~90% in some outbreaks - significantly higher than the ~50% fatality rate of two prevalent ebolavirus species, Ebola virus (EBOV) and Sudan virus (SUDV) 5-8.” References must be included that list the number of all previous cases of infection with EBOV, SUDV and Marburg viruses and the ensuing deaths.

Response: We included the number of deaths and reported cases for both EBOV and MBV. SUDV was not included, as the manuscript focuses on comparisons between EBOV and MBV. Additionally, we cited two CDC websites that provide the most up-to-date information on EBOV and MBV infections.

“The average case fatality rate of MBV infection is 73% (409 deaths among 563 reported human cases), markedly higher than that of Ebola virus (EBOV), a member of the Ebolavirus genus, which averages 44% (14,881 deaths among 33,820 cases) ^{1,2}.”

(11) Review: The text discusses N84 while the residue in the figure is labelled as N94.

Response: Thank you for catching this mistake. The correct residue is N94, and we have corrected this throughout the text and figures.

(12) Review: Please provide details in the methods section how recombinant NPC1 was prepared.

Response: We thank the reviewer for pointing this out. We have now added detailed methods describing the preparation of recombinant NPC1, which can be found in the revised Methods section under “Expression and purification of GP and NPC1”.

Reviewer 5

(1) Review: Little functional data is presented to buttress the new GPcl-NPC1 structure and the authors’ inferences from it regarding the mode of MARV GP-NPC1 interaction. At minimum, they should perform mutagenesis on the new contact residues in NPC1 they have identified and assess the binding of these mutants to EBOV and MARV GPcl.

Response: To directly address this concern, we have added several new lines of functional validation. These include::

- (i) Comparative receptor binding analysis. Using SPR, we directly compared the binding affinities of RAVV and EBOV GPcls, performing side-by-side measurements in triplicate for statistical robustness. RAVV GPcl bound NPC1-C ~11-fold more strongly than EBOV GPcl (Kd values of 394 nM and 4.34 μ M, respectively) (Fig. 3D, 3E).
- (ii) Mutational analysis of NPC1-C loop 3. To test the structural role of loop 3, we substituted Asp436 and Ile437 with alanine and measured GPcl binding by SPR. These substitutions markedly reduced RAVV GPcl affinity but had little effect on EBOV GPcl, confirming that loop 3 specifically stabilizes RAVV recognition (Figs. S7C, S9–S11).
- (iii) Mutational analysis of additional NPC1-C residues. We also introduced mutations at other NPC1-C residues that differentially interact with RAVV and EBOV GPcls, with results consistently supporting our structural analysis.

Conclusion: Together, these differences - particularly the new loop 3 contacts - explain why RAVV GPcl binds NPC1-C with substantially higher affinity than EBOV GPcl.

(2) Review: There is a body of literature from multiple groups describing MARV GP and NPC1 mutagenesis, the mode of GP-NPC1 interaction, NPC1 ortholog- and filovirus GP-dependent similarities and differences in this interaction for ebolaviruses and marburgviruses,—a partial list below: The authors should discuss their

structure in light of this work.

Lasso G et al., 2025, PMID: 39818205

Takada A et al., 2020, PMID: 31940478

Takada A et al., 2018, PMID: 30010949

Bornholdt Z et al., 2015, PMID: 26908579

Ng M et al., 2015, PMID: 26698106

Manicassamy B et al., 2007, PMID: 16989883

Response: We thank the reviewer for this thoughtful comment. We have revised the manuscript to include the following discussion:

“Previous mutagenesis studies have provided valuable insights into filovirus GP–NPC1 interactions, identifying numerous GPcl and NPC1 residues that influence binding. Two studies reported that NPC1 from an African fruit bat binds MBV GPcl but not EBOV GPcl, due to its Phe502 (an aspartate in human NPC1)^{27,28}. Our structure clarified this difference: in human NPC1, Asp502 forms a salt bridge with Lys155 in EBOV GPcl and does not contact the corresponding Leu139 in MBV GPcl (Fig. S13A). By contrast, Phe502 in bat NPC1 would clash with Lys155 in EBOV GPcl, restricting EBOV infection, while likely forming a favorable contact with Leu139 in MBV GPcl, thereby supporting MBV infection. A computational analysis suggested that Gly149 in EBOV GPcl, when mutated to a larger residue, would open the RBS and indirectly create new interactions with NPC1²⁹. Our structure confirmed this prediction: the corresponding Gln133 in MBV GP facilitates movement of the His131–Gln133 loop toward NPC1–C loop 3, establishing new contacts absent in EBOV GPcl (Fig. S13B). Another study reported that a P424A mutation in NPC1 impaired EBOV entry but not MBV entry³⁰, consistent with our structural finding that Pro424 interacts with EBOV GPcl but not with MBV GPcl (Fig. S6B). Additionally, one study found that R73A and K79A mutations in MBV GPcl abolished viral infectivity and suggested involvement in NPC1 binding³¹. However, our structure revealed that these residues do not contact NPC1; rather, they stabilize GP2 and the GP1/GP2 interface, respectively, thereby supporting proper folding. Indeed, the same study also showed that many conserved residues in filovirus GPs are essential for maintaining structural integrity³¹, consistent with our observations. Overall, our findings reconcile and clarify results from prior mutagenesis studies, providing broader insights into filovirus GP/NPC1 interactions.”

(3) Review: The claim that MARV GP mediates entry more efficiently than EBOV GP is unconvincing, especially given the data presented. For instance, the authors are looking at viral entry using a cell line—any differences they see could just be an artifact of the specific system they are using. If they believe that the higher GPcl–NPC1 binding affinity affords higher entry efficiency, they should support that hypothesis with data. For example, they could mutate MARV GP to reduce its affinity to a similar level as that of EBOV GP and then look at viral infectivity. It is likely that there is not a linear relationship between binding affinity and entry efficiency, especially given the likely avid interaction between virion-bound GPcl and NPC1 in late endosomes and lysosomes.

Response: To further establish that MBV GPs mediate entry more efficiently than EBOV GP, we expanded pseudovirus entry assays across three relevant human cell types (Huh7, HUVEC, and THP-1–derived macrophages), using a novel double-normalization method that enables direct comparison at equivalent expression levels. RAVV and Musoke pseudoviruses entered these cells far more efficiently than EBOV pseudoviruses (>300-, 25-, and 12-fold in three cell types, respectively) (Fig. 1D; Fig. S4). While we recognize that binding affinity alone may not linearly determine entry efficiency, our structural and functional data together—including higher NPC1 binding affinity, a distinct NPC1 binding mode, a partially flexible glycan

cap, and pronounced susceptibility to NPC1-triggered conformational changes—provide a coherent explanation for the enhanced entry mediated by MBV GPs.

(4) Review: The idea that the presented differences in entry affect virulence is even more of a stretch, given the many substantial biological differences between ebolaviruses and marburgviruses. In this reviewer's opinion, these experiments are tangential to the main thrust of the paper, which reports new structures. The authors would be better served focusing on the mechanism of the GP: NPC1 interaction and its direct biological implications.

Response: All comparisons of viral infectivity and virulence have been removed from the manuscript. The revised version now focuses exclusively on comparing the cell entry efficiencies mediated by MBV and EBOV GPs. We also explicitly acknowledge that many factors contribute to the high fatality rates of MBVs, with cell entry representing only one of them.

Responses to Reviewers' comments:

Reviewer 2

(1) Review: In the revision, the authors include tagged GPs to normalize pseudotyped viruses. They assume that by normalizing for expression levels, any observed differences in RUs must reflect differences in viral entry efficiency. However, expression levels alone do not necessarily correlate with production of functional pseudotyped viruses. Other mechanisms could also contribute to the observed phenotypes. For example, variations in the proportion of functional GPs produced in 293T cells, or differences in the efficiency with which these GPs are incorporated into viral particles. The authors should discuss these possibilities.

Response: We thank the reviewer for raising this point. To address the concern, we have revised the end of the pseudovirus entry section in the Results to read:

“The results showed that both wild-type RAVV and Musoke pseudoviruses entered all three human cell types far more efficiently than wild-type EBOV pseudoviruses (by more than 300-, 25-, and 12-fold in the respective cell types), demonstrating that MBV GP is substantially more effective at mediating viral entry than EBOV GP (Fig. 1D). Although these data support higher entry efficiency mediated by MBV GPs under conditions of matched total GP expression, our assay cannot exclude differences in the proportion of functional versus misfolded GP produced in HEK293T cells and incorporated into pseudovirus particles.”

(2) Review: The authors refer to K_d as affinity in Figure 2, while in Figure 3, they use K_a . They likely mean K_D , which is defined by k_d/k_a . The authors should clarify whether the data presented corresponds to association (k_a) or dissociation (k_d) rates, explain the rationale for using different terms or units across figures, and ensure consistent nomenclature throughout.

Response: In all cases, the values we intended to report are equilibrium dissociation constants (K_D) and, where specified, their reciprocal equilibrium association constants (K_A). We have revised the figures, legends, and manuscript text to use this notation consistently throughout.

Reviewer 4

(1) Review: The authors have adequately addressed most of my points. Considering that conclusions based on structural information should be confirmed by functional studies, it should also be examined whether mutating NPC1 residues D436 and I437 not only reduces MARV-GP but not EBOV-GP binding but also diminishes MARV-GP but not EBOV-GP-driven cell entry.

Response: We agree that testing MARV- and EBOV-GP-driven entry in cells expressing NPC1 D436 and I437 mutants would provide an additional functional complement to our structural work. In the current study, we have already performed quantitative SPR measurements showing that the D436A and I437A mutations markedly reduce RAVV GPcl binding while having minimal effect on EBOV GPcl binding (Extended Data Fig. 7d), in agreement with the RAVV-specific loop-3 contacts observed in our structures.

Implementing the proposed cell-entry experiments in a rigorously controlled manner would require generating NPC1-deficient cells and reconstituting them with wild-type or mutant NPC1, followed by

pseudovirus entry assays for multiple GPs. Importantly, analogous to our double-normalization strategy for GP, such experiments would also require normalization of *endosome-membrane levels of wild-type versus mutant NPC1*. Achieving this would necessitate NPC1-directed neutralizing sera or other quantitative reagents, which we do not currently have. Without reliable normalization of functional NPC1 expression, reduced entry could not be unambiguously attributed to altered binding rather than to differences in receptor abundance or processing. For these reasons, we feel that a full set of pseudovirus entry assays with NPC1 mutants constitutes a substantial new line of work beyond the scope and timeline of the present study.

In light of comments from the reviewers and editor, we have tempered our interpretation of loop-3 contributions to MBV entry. Throughout the manuscript, we now describe D436 and I437 and the associated loop-3 contacts as *contributors, rather than critical contributors*, to MBV GP–NPC1 recognition, and explicitly acknowledge that additional NPC1–GP interactions and other entry mechanisms also contribute to MBV entry. We hope these revisions appropriately qualify our conclusions while leaving this as a direction for future work.

Reviewer 5

(1) Review: Overall, the most interesting and important aspect of this work, in my opinion, is the evidence for a novel mode of NPC1 binding by MARV GPcl relative to EBOV GPcl. Please see below for some specific comments related to this. I feel that this is the real strength of this manuscript. By comparison, the structure with the VHH, while interesting, is less noteworthy to me. Thus, I am also less concerned about the *in vivo* data with the VHH (or lack thereof).

Response: We thank the reviewer for these positive comments and for highlighting the novel NPC1–MBV GPcl binding mode as the main strength of our study. Consistent with this assessment and with the editor’s request to shorten the manuscript, we have de-emphasized the nanobody component and sharpened the focus on NPC1-mediated entry.

Specifically, we have (i) removed the nanobody from the manuscript title (which also satisfies the 75-character limit), (ii) deleted the paragraph on the nanobody from the Introduction, and (iii) shortened the nanobody Results section by removing the detailed comparison between Nanosota-MB1 and two previously described human antibodies. We now present Nanosota-MB1 primarily as a proof-of-concept neutralizing nanobody and an additional structural probe, while emphasizing that the central conceptual advance of the study is the distinct NPC1 binding mode of MBV GPcl.

(2) Review: To me, the claim that the enhanced binding affinity of MARV GPcl to NPC1-C accounts for its increased entry activity still seems like a reach given the complexity of the biology at both the single-cell and organismal level. I feel the authors' emphasis of this points detracts from the manuscript. In my opinion, the conclusions from these experiments should be greatly de-emphasized.

Response: We agree that MBV entry is governed by multiple factors beyond NPC1 binding affinity alone. Our intention was not to imply that the enhanced affinity of MBV GPcl for NPC1-C fully accounts for the increased entry activity, but rather that it represents one contributing component suggested by our structural and binding data. We now explicitly note in the Discussion that, although a positive relationship between receptor affinity

and entry efficiency has been clearly established for some other virus families (e.g., coronaviruses), this relationship is less well defined for filoviruses due to the complexity of the filovirus entry pathway.

In addition, the final paragraph of the Discussion emphasizes that several structural features, including the partially flexible glycan cap, distinct NPC1 binding mode, increased NPC1-binding affinity, and pronounced NPC1-triggered conformational changes, are all likely to contribute, in combination, to enhanced MBV entry.

We hope this clarifies that we view increased NPC1 affinity as one contributor among several mechanisms that may underlie the higher entry efficiency of MBVs, rather than as a standalone explanation for this complex phenotype.

(3) Review: Lines 165-169: To facilitate readership, please specify the residue numbers comprising each NPC1 loop when first describing the loops mediating the interaction with RAVV GPcl.

Response: We have now specified the residue ranges for each NPC1-C loop at their first mention in the Results section. The relevant sentence has been revised to:

“Whereas NPC1-C uses two protruding loops (loop 1, residues 418–425, and loop 2, residues 498–508) to bind the EBOV RBS, it employs three loops, including an additional loop 3 (residues 436–437), to bind the RAVV RBS (Fig. 3c, 3d).”

(4) Review: Lines 180-183: Mutagenesis experiments show that residues in loop 3 are important but I would not say they are critical since the mutant NPC1 still binds (K_a increases by less than a log). I suggest lowering the tone of the statement.

Response: We have softened the wording in the Results. The relevant sentence now reads: “Alanine substitutions at these two positions markedly reduced RAVV GPcl binding affinity (Extended Data Fig. 7d; Supplementary Information Fig. 2), confirming that loop 3 contributes to RAVV recognition.”

(5) Review: Line 205-207: K155 and D502 are too far (4.7Å) to form a salt-bridge in EBOV GP – NPC1 complex (pdb: 5f1b). Also, F502 in NPC1 doesn't seem to clash with K155 in EBOV GP if I perform in-silico mutational analysis on the EBOV GP – NPC1 complex. I would suggest revisiting this section of the ms and include intermolecular distance measures in the corresponding figures. Are L139 and F205 to form a favorable hydrophobic contact?

Response: We agree with the reviewer that these specific interactions, which were described and discussed in previously published EBOV GP–NPC1 structural analyses rather than derived from our own data, are weak and not essential for the main conclusions of our study. In line with the editor's request to shorten the manuscript and to avoid over-interpreting interaction details that we did not directly verify, we have removed this passage from the Results and no longer discuss these particular contacts. The corresponding text and figure legend have been simplified accordingly.

(6) Review: Line 210: What is the area of the binding pocket in RAVV GPcl compared to EBOV GPcl? This would help further supporting the idea that Q133 contributes to the opening of the pocket. However, it is

possible that other residues differences contribute too (e.g. the lack of disulfide bond between 121-147 cysteines -EBOV numbering). I strongly suggest that the authors compare the geometry of the GP1 pocket in EBOV and RAVV. Along these lines, the positioning of the beta sheets on one side of the RAVV GP1 (residues 92-98 & 121-128) pocket is remarkably different from that in EBOV GP1, narrowing the opening on the pocket. I didn't see any comment on this in the manuscript.

Response: We agree that comparing GP1 pocket geometry is important for understanding how Q133 and neighboring residues contribute to NPC1 engagement. In the revised Results, we expanded our structural description of the pocket. The new text reads:

“To further understand the distinct mode of NPC1 binding by RAVV GP1, we compared the shapes of the RBS pocket openings in the RAVV and EBOV GP1s (Fig. 3f). Relative to EBOV GP1, three regions surrounding the RAVV RBS pocket undergo pronounced conformational changes. First, residues 130–133 swing outward, a rearrangement likely facilitated by the loss of the EBOV disulfide bond (Fig. 3f). Moreover, previous computational analysis suggested that mutation of EBOV GP1 Gly149 to a bulkier residue would open the RBS (pushing the opening outward) and indirectly create new interactions with NPC1. Our structure confirms this prediction: the corresponding Gln133 in MBV GP promotes outward movement of the Pro130–Gln133 loop toward NPC1-C loop 3, forming new contacts that are absent in EBOV GP1 (Extended Data Fig. 8b). The other two changes involve residues 62–65 and 124–128; extensive residue substitutions in these segments drive them outward and inward, respectively (Fig. 3f). Collectively, the reshaped RBS pocket openings in RAVV GP1 create additional NPC1 contacts and compel NPC1-C to bind in a distinct, higher-affinity orientation.”

The segments 92–98 and 121–128 highlighted by the reviewer form part of these rearranged GP1 elements and contribute to the altered contour of the pocket opening in RAVV compared with EBOV. Although we removed some peripheral literature discussion to streamline the manuscript, we explicitly acknowledge the prior computational prediction regarding EBOV Gly149. Our structure is fully consistent with that prediction and further shows how additional residue differences, including the absence of the Cys121–Cys147 disulfide, remodel the GP1 pocket. *Together, these earlier computational studies provide strong support for the structural observations reported here.*

(7) Review: Could the authors speculate in their discussion what factors might drive the different binding mode of RAVV GP1? Is the shape of the RAVV GP binding pocket driving the binding mode? This is to me a very interesting finding that would benefit from some additional explanation.

Response: We thank the reviewer for this thoughtful observation and comment, which helped guide our revised discussion of the determinants of the distinct NPC1 binding mode. As noted in our response to Comment #6, we have expanded the Results section to compare the geometry of the GP1 pocket in RAVV and EBOV GP1 and to describe how conformational changes in residues 62–65, 92–98, 121–128, and 130–133, together with Gln133 and the absence of the Cys121–Cys147 disulfide, reshape the RAVV RBS pocket. These features create additional NPC1 contacts and help explain why NPC1 binds RAVV GP1 in a different orientation and with higher affinity than EBOV GP1.

(8) Review: Lines 292-294. The sentence implies that loop 3 is particularly relevant for the ~11-fold higher binding affinity to human NPC1 than EBOV GP1. However, the experimental data suggest that the interactions

driven by H418 and I419 are more relevant. These mutagenesis experiments assess the individual contribution of each residue separately, but it does not address potential epistatic effects. I would suggest caution when describing the importance of loop 3. While important, the experimental data does not support that these are critical residues for binding.

Response: We agree that our original wording overstated the specific contribution of loop 3 relative to other NPC1–RAVV contacts. In the revised manuscript, we have removed the phrase “particularly new contacts between NPC1-C loop 3 and the RAVV RBS” and now describe the higher affinity of RAVV GPcI for NPC1-C as arising from multiple structural differences between the RAVV and EBOV interfaces. The relevant sentence now reads:

“Third, sequence divergence between the RAVV and EBOV RBS generates markedly different interaction networks and reshapes the openings of the RBS pocket. Finally, these differences enable RAVV GPcI to bind NPC1-C with ~11-fold higher affinity than EBOV GPcI.”

Elsewhere, we refer to loop 3 as one of several contributors to RAVV recognition rather than as a critical determinant, which is more consistent with the mutagenesis data and the reviewer’s suggestion.

(9) Review: Include distances in the figures that describe molecular interactions: Figs 5b, S7, S8, S12 & S13.

Response: We have revised these figures and the corresponding extended data figures (some of which were reorganized to comply with the journal’s limits on extended data items) to include intermolecular distance measurements for the key side-chain interactions described in the text.